# Altruistic disease signalling in ant colonies

Erika H. Dawson [1,2] ✉, Michaela Hoenigsberger[1,5], Niklas Kampleitner[1,5], Anna V. Grasse [1], Lukas Lindorfer [1], Jennifer Robb[1,4], Farnaz Beikzadeh[1], Florian Strahodinsky[1], Hanna Leitner[1], Harikrishnan Rajendran[1], Thomas Schmitt [3] & Sylvia Cremer [1] ✉

Sick individuals often conceal their disease status to group members, thereby preventing social exclusion or aggression. Here we show by behavioural, chemical, immunological and infection load analyses that sick ant pupae instead actively emit a chemical signal that in itself is sufficient to trigger their own destruction by colony members. In our experiments, this altruistic disease-signalling was performed only by worker but not queen pupae. The lack of signalling by queen pupae did not constitute cheating behaviour, but reflected their superior immune capabilities. Worker pupae suffered from extensive pathogen replication whereas queen pupae were able to restrain their infection. Our data suggest the evolution of a finely-tuned signalling system in which it is not the induction of an individual's immune response, but rather its failure to overcome the infection, that triggers pupal signalling for sacrifice. This demonstrates a balanced interplay between individual and social immunity that efficiently achieves whole-colony health.

Sick animals living in groups are often excluded or subjected to aggression by other group members, both to reduce the risk of infection and to gain social or resource-related advantages over the weakened individual[1,2]. As a result, these individuals often conceal their infection status by, for example, suppressing sickness behaviours in the presence of conspecifics[3,4]. This is not the case when group members are kin, since relatedness diminishes the conflict of interest between sick and healthy individuals. Therefore, even when group members can detect disease in others, for example, using sickness cues such as altered physical appearance or behaviour, avoidance is often only displayed against non-kin, whilst normal social interactions are maintained with diseased kin[5].

Diseased individuals can therefore benefit, not only by not concealing, but even from actively communicating their health state to their relatives in order to receive additional care. As such, fungal pathogen-exposed termites display a vibratory signal[6], inducing grooming behaviour by nestmates[7] and wounded ants produce behavioural and chemical displays attracting their colony members to care for their wounds[8,9], which in both cases reduces the signalling individual's infection risk and hence improves its survival. Helping a relative to survive infection can also indirectly benefit the caregiver by

increasing its own inclusive fitness via shared genes with the recipient[10]. Active disease-state signalling is therefore selected for in groups of relatives, when both the signaller and the signal-receiver gain from the care response. But what happens when the related group member responds not with care, but instead with exclusion or aggression toward the infected individual to protect the group? Should the infected individual still signal its sickness to others, even at the risk of being sacrificed for the benefit of the group?

The evolution of such 'altruistic signalling' of infectious disease should be promoted in social groups by two factors: (i) high relatedness, leading to a large indirect fitness gain to the sacrificed individual, if signalling promotes group health, and (ii) low direct fitness loss to the sacrificed signaller, i.e., when its expected future reproductive value is small. Both are fulfilled in the non-reproductive workers of social insect colonies. Workers are typically highly related with one another and produce no offspring of their own, but instead rear the queen's brood and maintain the colony[11]. Therefore, the fitness of workers depends on the fate of the colony as a whole. Here, we experimentally test whether altruistic signalling of one's own sickness indeed evolved in social insect colonies.

[1]ISTA (Institute of Science and Technology Austria), Klosterneuburg, Austria. [2]Université Sorbonne Paris Nord, Villetaneuse, France. [3]University of Würzburg, Würzburg, Germany. [4]Present address: Technical University Munich, Munich, Germany. [5]These authors contributed equally: Michaela Hoenigsberger, Niklas Kampleitner. ✉e-mail: dawson.erika.h@gmail.com; sylvia.cremer@ist.ac.at

Social insects, particularly termites, ants and social bees, show sophisticated collective disease defences when colony members come into contact with pathogens. These social immunity measures range from sanitary care of individuals that can still be rescued, to elimination of fatally-infected colony members[12,13]. The latter can occur via 'hygienic cannibalism' as observed in termites[14,15] and in ant queens during their nutrient-deprived colony founding phase[16]. In bees[17] and ants[18], it often involves the removal of typically immobile larvae or pupae from the nest, while the mobile adults often leave the colony when approaching death[19–21]. Here we study the invasive garden ant, *Lasius neglectus*, in which adult workers care for the brood, yet switch to destructive disinfection of worker pupae that suffer from deadly infections of the fungal pathogen *Metarhizium brunneum*[22]. Since pupae of this ant species are enclosed in a cocoon, the first step in this multicomponent behaviour is that workers prematurely unpack them from their cocoon, followed by biting and disinfection. This process prevents pathogen replication in the host and ultimately limits the spread throughout the colony. Notably, this destructive disinfection is performed by the workers during the non-infectious incubation period of the fungal pathogen, therefore not inducing any disease risk to the workers[22]. The expression of this behaviour depends on detecting disease-related changes in the pupal surface chemistry. In particular, unpacked pupae showed increased abundance of four cuticular hydrocarbon (CHC) peaks (tritriacontadiene, C33:2; tritriacontene, C33:1; pentatriacontadiene, either alone as C35:2, or co-eluting with pentatriacontene, C35:2 + C35:1), and experimental CHC removal prevented unpacking behaviour[22]. Yet, it remains unknown whether these chemical changes are simply passive cues resulting from infection or an induced immune response, or whether they represent active signals produced by infected pupae to trigger their own destruction for the colony's benefit – thereby increasing their inclusive fitness. In this study, we manipulate the context in which pupae could signal their disease status, enabling us to disentangle these two scenarios.

## Results and discussion

Individual pupae were experimentally infected with the pathogen or sham-treated and then kept either with two workers or alone. We employed this full-factorial approach as we speculated that if a signal is costly to produce (e.g.,[23]) and pupae are able to sense the presence or absence of workers, then they should only signal when both infected and in the presence of the intended receivers, i.e., the workers. As close social interactions in ant colonies often involve the transfer of CHCs through grooming and food sharing – processes that contribute to the colony-wide "gestalt" odour[24,25] – pupae, which do not feed at this developmental stage, likely acquire CHCs from workers via external contact. Since the CHC profiles of *L. neglectus* workers[26,27] and pupae[22] overlap, we therefore needed to disentangle pupae- and worker-derived chemical compounds. To exclusively quantify the pupa-produced CHCs, we used stable isotope labelling to enrich only the workers' but not the pupal CHCs with the natural carbon isotope $^{13}C$ by feeding the workers $^{13}C$ glucose before the experiment (as detailed in the "Methods", Supplementary Fig. S1). We focused our analysis on the four previously identified candidate peaks for possible unpacking cues/signals, with two of them (tritriacontadiene, C33:2, and tritriacontene, C33:1) in particular being associated with immune system activation[22].

### Worker pupae signal their infection in worker presence

Our chemical analysis by use of gas chromatography–mass spectrometry (GC–MS) clearly revealed that only the two immune-associated peaks C33:2 and C33:1 were upregulated, and only by worker pupae that were both infected and kept with workers (Supplementary Fig. S2a; [LM] infection treatment*worker presence interaction: C33:2: $p = 0.008$, C33:1: $p = 0.036$; Supplementary Table S1a), such that their chemical profile deviated from that of the other three groups (Fig. 1a,

Supplementary Table S2; all post hoc *p*-values to the other three groups < 0.05). The fact that only the two immune-associated peaks were increased in abundance by infected pupae with workers, and that infection load did not predict abundance of either peak (both $p > 0.16$; Supplementary Table S1b), provides double evidence that this upregulation is not directly brought about by the fungus itself.

Although this CHC change only occurs in infected and hence immune-activated ants, our data reveal that it is not simply a secondary effect of the immune response. The expression patterns of our three tested candidate genes, which represent the three levels of the immune response[28] – pathogen recognition (β-1,3-glucan binding protein, *BGBP*[29]), immune regulation (peptidoglycan recognition protein SC2, *PGRP-SC2*[30,31]) and the immune effector activity (Defensin[32], *Def1*; see Methods) – were consistently upregulated in response to infection, while worker presence had no effect (Fig. 1c and Supplementary Fig. S3a [LM] infection treatment: all genes $p < 0.001$; worker presence: all genes $p > 0.15$; Supplementary Table S3a). Therefore, even if all infected pupae equally upregulate their immune gene expression above the healthy baseline, only those in the presence of workers also show an increase in specific CHCs. This indicates that CHC upregulation must be under dynamic host control, allowing the pupae to actively signal their disease status.

### Queen pupae do not signal but can restrain infection

While altruistic disease signalling is expected to evolve in sterile workers, the situation is more complex for the daughter queens. By alerting others to destroy them, queen pupae would risk losing potential future reproductive opportunities if they would survive infection. On the other hand, by spreading their infection to their colony, they could incur high indirect fitness costs. We therefore tested if infected *L. neglectus* queen pupae exhibit altruistic disease signalling similar to that of the worker pupae. We found that – contrary to the worker pupae – none of the four candidate peaks (Fig. 1b and Supplementary Fig. S2b; [LM] all *p*-values > 0.82, Supplementary Tables S4a, S5) nor any of their other CHC peaks (see "Methods"; Supplementary Table S6, [LM] all *p*-values > 0.14) showed differences between the four treatment groups. Thus, we found no evidence of disease signalling in the queen pupae, despite their exposure to a considerably higher pathogen dose than the worker pupae (see "Methods"). Consequently, whilst the signalling infected worker pupae elicited destructive disinfection by the workers, who unpacked them above baseline levels (Cox proportional-hazards regression, treatment effect $p = 0.0037$; Supplementary Table S7), this was not the case for the non-signalling infected queen pupae ($p = 0.115$; Supplementary Table S7). Despite these differences in CHCs between worker and queen pupae, the immune responses of the two castes showed the same pattern. Also in the queen pupae, we found the expression levels of all our candidate genes to be upregulated upon infection, whilst being independent of worker presence (Fig. 1d and Supplementary Fig. S3b [LM] infection treatment: all genes $p \leq 0.001$; worker presence: all genes $p \geq 0.15$; Supplementary Table S8a). Notably, however, the baseline immune investment under healthy conditions was 35% higher in queen pupae than in worker pupae (Wilcoxon Rank-Sum test; healthy pupae: $W = 846$, $p < 0.0001$), whereas both castes showed similar expression levels when infected ($W = 1281$, $p = 0.387$). This reveals that the documented higher immunocompetence of the queen caste in social insects[33,34] already emerges at the brood stage when, as previously shown in other insects, immune activity can already be quite pronounced[35,36] despite the fact that pupae do not feed and undergo complete transformation during metamorphosis. Higher baseline immune investment is known to reduce the likelihood of disease development[37,38], which may enable queen pupae to better cope with infection compared to worker pupae.

We therefore asked if the absence of signalling in the queen pupae represents a selfish behaviour which puts the colony at risk, or is an

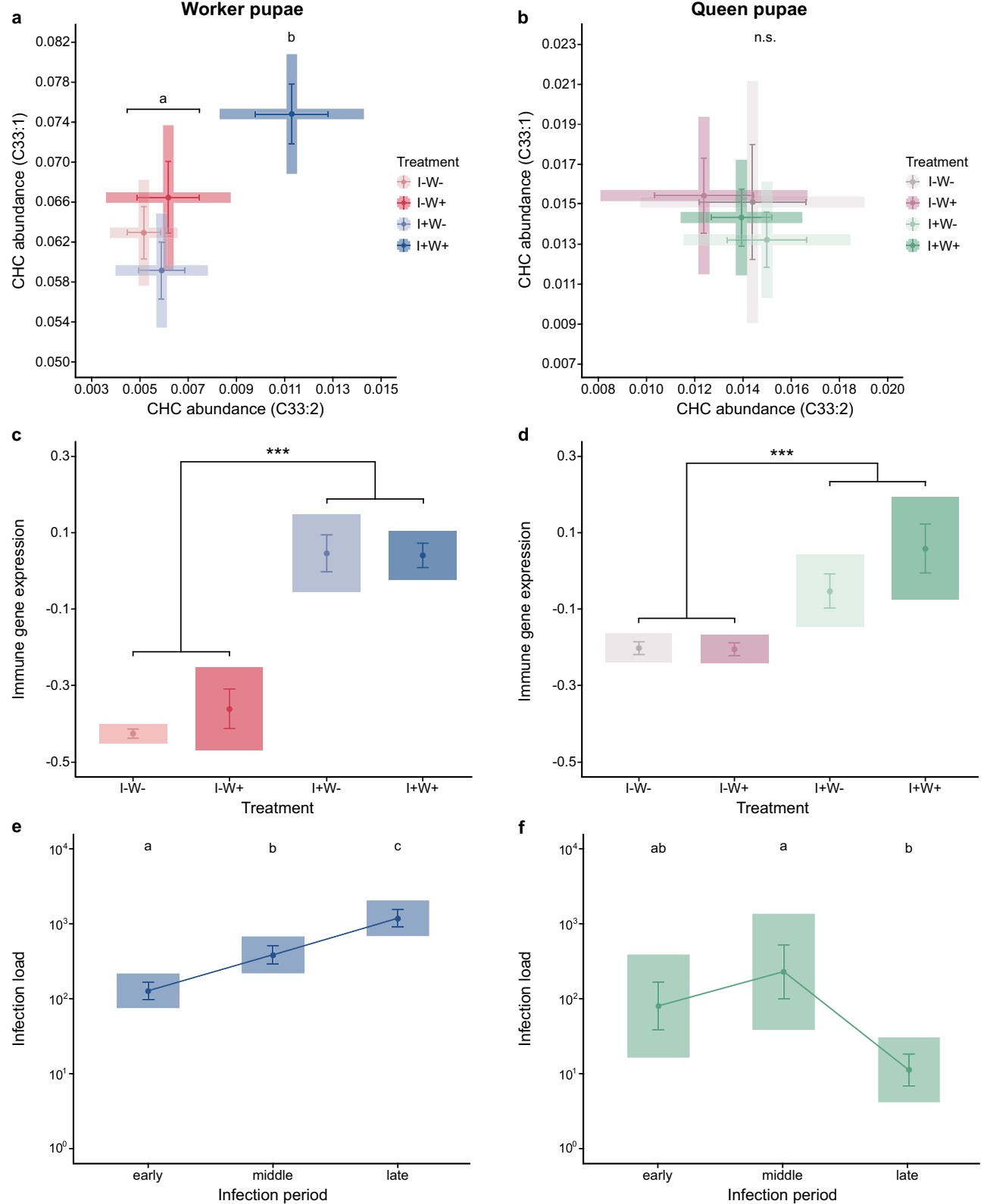

honest representation of their higher immunocompetence, allowing them to cope with infection and making the need for signalling obsolete. To this end, we compared pathogen load progression in queen and worker pupae. We found that, whilst infection load steadily increased over the course of infection – with no sign of even plateauing – in the worker pupae (Fig. 1e; all *p*-values between early, middle and late infection period *p* < 0.0158; Supplementary Table S9a), queen

pupal infection load peaked intermediately, but then decreased by almost threefold (Fig. 1f; middle to late infection period *p* = 0.006, Supplementary Table S9b). This suggests that, since the stronger queen immune system was able to reduce infection, whilst the less potent worker immune system was not, both the signalling in the worker pupae and the absence thereof in the queen pupae are honest reflections of the caste-specific differences in individual disease

**Fig. 1 | Chemical signalling, immune gene expression and infection load of ant pupae. a** Infected worker pupae (I+, blue) showed upregulated relative abundance of cuticular hydrocarbons tritriacontadiene (C33:2) and tritriacontene (C33:1) under worker presence (W+; dark tone) only. Their chemical profile differed from infected pupae in the absence of workers (W-; pale tone; LMM, pairwise-post hoc corrected, two-sided p-values, Supplementary Table S2: $p = 0.0001$), and control pupae with (I-W+; dark red; $p = 0.012$) and without workers (I-W-; pale red; $p = 0.002$; significance groups denoted as a, b). **b** Queen pupae showed no difference between the four treatment groups (infection green, sham purple; LMM, overall model uncorrected, two-sided $p = 0.434$, n.s., Supplementary Table S5). Pupal immune gene expression (combined analysis of the three candidate genes *BGBP*, *PGRP-SC2* and *Def1*), in contrast, was increased in both worker pupae (**c**) and queen pupae (**d**) only in response to infection (LM; main effects uncorrected, two-sided p-values worker pupae: $p = 2.2 \times 10^{-16}$, queen pupae: $p = 0.0003$, depicted as***; Supplementary Tables S3b, S8b), but not to worker presence (worker pupae:

$p = 0.467$, queen pupae: $p = 0.338$). Pupal infection load (given as fold-change to the maximum caste-specific exposure dose) of (**e**) worker pupae increased steadily over time (LM, pairwise-post hoc corrected, two-sided $p$-values, early-mid: $p = 0.016$, mid-late: $p = 0.0007$, early-late: $p = 7.2 \times 10^{-9}$; Supplementary Table S9a), whilst in queen pupae (**f**) infection was reduced back to early infection levels over the course of the experiment (LM; pairwise-post hoc corrected, two-sided p-values, early-mid: $p = 0.016$, mid-late: $p = 0.006$, early-late: $p = 0.180$; Supplementary Table S9b, significance groups denoted as a–c). Graphs show means ± sem as dots and whiskers in shaded 95% CI box, with data $\log_{10}(x+1)$-transformed in (**c–f**). Chemical analysis based on 426 pupae, with the subset of 265 infected pupae analysed for infection load; immune gene expression based on 179 pupae. See Supplementary Fig. S2 (Supplementary Tables S1a,S4a,S6) for individual CHCs, and Supplementary Fig. S3 (Supplementary Tables S3a,S8a) for individual immune genes. Source data (1a–f) are provided as a Source Data file, raw data under https://doi.org/10.15479/AT-ISTA-20471.

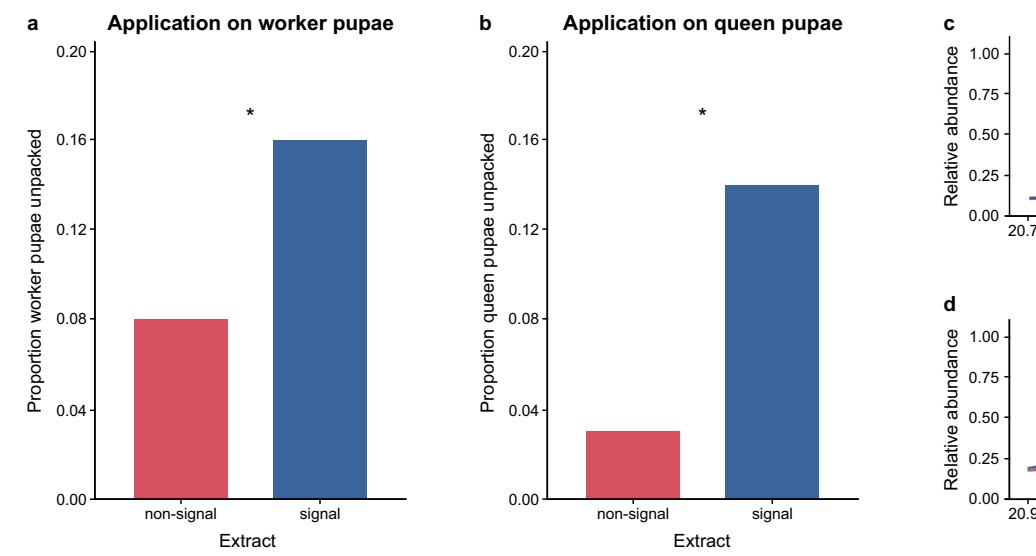

**Fig. 2 | Unpacking elicitation and chemical profiles of the CHC signal compounds.** Transfer of the CHC extracts of signalling pupae (blue) onto healthy (**a**) worker and (**b**) queen pupae was sufficient to elicit a higher proportion of unpacking than application of the non-signal extract in the bioassay (red; binomial GLMs, main effect of extract type, uncorrected, one-sided p-values testing for a higher unpacking induction by application of signal than non-signal extract onto worker pupae: $p = 0.0496$, resp. onto queen pupae: $p = 0.0145$; Supplementary Table S10). Chromatograms of the signal and the non-signal extract, showing the relative abundance (normalised to an internal standard and max-scaled) for (**c**) tritriacontadiene (C33:2), and (**d**) tritriacontene (C33:1), based on SIM traces of

their molecular ions. Overlays visualise the increase in the C33:2 and C33:1 hydrocarbon peaks in signalling compared to control pupae, as well as a change in isomeric composition in C33:1, revealed by its multi-shoulder peak patterns. Further analysis of the DMDS-derivatised extracts revealed at least eight positional C33:2 isomers and six positional C33:1 isomers (Supplementary Tables S11,S12 and Supplementary Figs. S4, S5), all with an inferred (*Z*)- resp. (*Z,Z*)-configuration (Supplementary Figs. S4, S6). Unpacking bioassay (**a**, **b**) based on 213 pupae, extract analysis (**c**, **d**) on a pool of 20 pupae each. Source data (2**a**, **b**) are provided as a Source Data file.

progression during our experiment. Future studies could identify under what infection conditions individual immunity of the queen pupae may fail, and whether this would lead to signalling for own destruction also in the queen caste. We expect this to happen only when pupal immunity is overwhelmed. Our findings show that pupae do not signal the presence of infection nor the activation of their immune system, allowing them the opportunity to clear infections on their own before signalling. This implies the evolution of an early warning system, where fatally infected pupae signal to enable a rapid and targeted colony response.

### Signalling relies on complex CHC changes

To confirm a causal—not just correlational—relationship, we performed a bioassay to test whether applying chemical extract from signalling pupae to healthy brood would trigger worker unpacking, using extract from non-signalling pupae as a control (Methods). We indeed found that healthy worker pupae experimentally coated with the extracts of signalling pupae were unpacked more often than those

treated with non-signal extract (Fig. 2a; [GLM] one-sided $p = 0.0496$; Supplementary Table S10). Notably, transfer of the signal extract across castes, to healthy queen pupae, equally induced unpacking (Fig. 2b; one-sided $p = 0.0145$; Supplementary Table S10). This allows us to speculate that if queen pupae were to signal any fatal infection that they may develop, workers would respond by unpacking them.

Our bioassay demonstrated that workers rely on the distinct chemical profile of signalling pupae to initiate unpacking. Comparison of the C33:2 (Fig. 2c) and C33:1 (Fig. 2d) GC–MS chromatograms from signal and non-signal extracts indicated that these differences involve not only changes in overall peak intensity (for both C33:2 and C33:1), but also differences in peak shapes (particularly for C33:1) suggesting a possible variation in the isomeric composition of co-eluting positional isomers of the same chain length, bond type and number, but differing in bond position[39]. Indeed, five isomers of C33:1 were previously found to co-occur in *L. neglectus* workers, and the presence of several, as yet unspecified, isomers of C33:2 has also been suggested[26]. Isomeric characterisation using dimethyl disulfide derivatisation (DMDS,

Methods), revealed eight positional isomers of C33:2 (Supplementary Table S11 and Supplementary Fig. S4a) and six of C33:1 (Supplementary Table S12 and Supplementary Fig. S4b) in our pupal extracts.

While the overall abundance of the C33:2 isomers (dominated by 11,21-C33:2 and 10,20-C33:2; Supplementary Figs. S4a, S5a) was ~ 20-times higher in signalling compared to non-signalling pupae, no considerable shift in isomeric composition was detected. For example, 11,21-C33:2, made up 31% of the sum of all eight isomers in non-signalling pupae and 35% in signalling pupae, while 10,20-C33:2 accounted for 21% and 23% respectively (for details see Supplementary Table S11). The C33:1 isomers, on the other hand, while only 3-times more abundant in the signalling pupae, showed considerable proportional changes. In particular, the most prominent isomer, 12-C33:1, exhibited a striking reduction in proportion from 50% in non-signalling pupae to 31% in signalling pupae. In contrast, 13-C33:1 increased from 9% in non-signalling to 16% in signalling pupae, and 11-C33:1 doubled from 9% to 18% (Supplementary Table S12). The remaining C33:1 isomers did not show any substantial differences between signalling and non-signalling pupae. Notably, 10-C33:1 and 9-C33:1 are part of the isomer mixture contributing to the previously identified C33:1 signal peak[22], while 7-C33:1 constitutes a separate C33:1 peak that is not upregulated in unpacked pupae[22] (Supplementary Table S13, Methods). We therefore found that the chemical profile change in signalling pupae is characterised by a complex interplay: an overall increase in the relative abundance of both C33:2 and C33:1 on the overall pupal bouquet, alongside distinct up-and-downregulation of specific C33:1 isomers. It remains unresolved whether ant workers react to the combined effect of multiple components in this composite pattern, or primarily to a single isomer whose abundance changes.

Evidence for both comes from honeybees, where hygienic behaviour – the uncapping of the brood cell and removal of diseased brood[17]– can be triggered either by the experimental application of single synthetic isomers onto healthy brood alone[40], or by a bouquet of multiple compounds[41]. In addition, shifts in compound ratios have been shown to affect worker responses[42]. Notably, both C33:2 and C33:1 – described here as active signals of fungal disease in ants – have previously been found to be associated with unhealthy honeybees, although it is unclear whether in bees they also function as active signals or merely reflect a passive side effect of infection. Adult bees can show increased C33:2 following viral infection[43] and elevated C33:1 after immune-stimulation with bacterial cell wall components[44]. In honeybee brood, C33:1 is increased across different – viral and mite – infections[40,45]. While the isomeric composition is still unresolved for most of these infections, the (Z)-10-C33:1 isomer has been identified as a key compound upregulated in virally-infected honeybee brood and is a strong trigger of hygienic behaviour when experimentally applied in its synthesised form[40–42]. Interestingly, 10-C33:1 is also part of the isomer mixture of the C33:1 signal peak in *Lasius* ants, and comparison of the retention time of synthetic (Z)-10-C33:1 with our natural extracts revealed that the ants' 10-C33:1 isomer is present in the same (Z)-stereo-configuration as in the honeybee[40] (Supplementary Fig. S6). Since the (Z)-configuration is the only naturally-occurring stereoisomeric form of insect CHCs identified to date[46], and is thought to result from their biosynthetic pathways[47], it is highly probable that all identified C33:2 and C33:1 isomers found here represent the (Z)-form. This is biologically relevant given that stereoisomers can differ greatly in their biological activity[48]. Having confirmed the isomeric identity of the 10-C33:1 in bees and ants, we tested whether the synthetic (Z)-10-C33:1 in its pure form, shown to induce hygienic behaviour when applied to honeybee brood[40,41], would similarly elicit destructive disinfection in ants. However, it failed to trigger any unpacking in our bioassay (0/41 pupae unpacked; see Methods). This may not be surprising, given that (Z)-10-C33:1 was the least abundant of the isomers in the pupal C33:1 signal peak, comprising only 8% and showing no change in

proportion upon signalling (signal to non-signal ratio: 1.00; Supplementary Table S12).

Therefore, despite the overall similarities between bees and ants, the specific isomers acting as indicators of disease are system-specific, suggesting there is no universal disease marker across social insects. It is intriguing, however, that CHCs of similar chain length and with one or two double bonds play an important role in detecting disease states across diverse infections in both ants and bees, suggesting these insects primarily rely on unsaturated CHCs for disease detection. In contrast, sick termites, even when infected with the same fungal pathogen as the ants in our study, show alterations in methyl-branched alkanes[49]. Unsaturated and methyl-branched CHCs are generally considered to convey more information than saturated CHCs[50,51], which may explain their role in disease detection and communication. The differences between termites and social Hymenoptera (bees, wasps, ants) suggest that these two evolutionary distinct lineages of social insects may have evolved different pathways to detect disease, potentially driven by distinct biosynthetic pathways or sensory systems involved in producing or perceiving signs of disease[52]. Yet, the parallel evolution of informative disease signatures across the social insects highlights the fundamental importance of early disease detection for efficient colony-level disease defence.

Here we describe that ant pupae actively signal their disease to the colony, analogous to how infected cells in a multicellular organism alert immune cells to eliminate them, a self-sacrificial act that preserves the integrity of the whole[53]. This form of hygienic altruism demonstrates that mechanisms of immune-like defence are not restricted to within individual organisms but can also evolve between individuals in superorganismal colonies, where the fitness of each member depends on the health and success of the reproductive unit they collectively comprise[54–56]. The signal involves a complex change in the chemical profile produced by pupae when overwhelmed by infection, suggesting a finely tuned balance between individual and social immunity in social insect colonies. Universal destruction of all infected colony members based on passive cues of infection or immune system upregulation would instead disregard varying levels of immunity and recovery capabilities, potentially causing substantial, avoidable losses, including costly-to-rear individuals like new queens. Instead, a mechanism for individual health assessment, where individuals signal for destruction only when individual immunity fails, could allow for the simultaneous protection of both individual colony members and overall colony health. That this hinges on precise and complex chemical signalling, rather than broad-spectrum sickness cues, underscores the evolutionary refinement of social immunity systems in distinguishing between recoverable and terminally ill individuals.

## Methods
### Ant host
As host species, we used the invasive garden ant, *Lasius neglectus*. We collected several hundred workers, multiple queens and brood of this species from its introduced supercolonial populations[26,27] in Jena, Germany (N 50° 55' 54.599" E 11°35' 8.401") between 2018 and 2024, and in Seva, Spain (N 41°48' 32.699' E 2° 15' 42.3") between 2016 and 2024. Collection of this unprotected species from the field was performed in compliance with international regulations, such as the Convention on Biological Diversity and the Nagoya Protocol on Access and Benefit-Sharing (ABS; permit numbers: ESNC12, ESNC126, SF/0558-0561, EPI 92-2023). After collection, ants were reared in large stock colonies in the laboratory with 30% sugar water and minced cockroaches. Experiments were performed with pupae of a standardised developmental stage (white pupae with black eyes, as visible through the cocoon using a stereo microscope (Leica S6 E) for worker pupae; for queen pupae, the stage can be reliable inferred by checking some pupae in the colony, as they develop highly synchronised in same-age

cohorts). Workers were always sampled from the same nest as the respective pupae from the inside of the brood chamber to avoid foragers. All experimental work followed European and Austrian law and institutional ethical guidelines.

## Fungal pathogen

As the pathogen, we used the entomopathogenic fungus *Metarhizium*, whose infectious conidiospores naturally infect ants[57] by penetrating their cuticle, killing them and growing out to produce highly-infectious sporulating cadavers[58]. We infected the pupae with conidiospores (abbreviated as 'spores' throughout) of *M. brunneum* strain Ma275 (KVL 04-57, obtained from J. Eilenberg and N.V. Meyling from the University of Copenhagen, Denmark), which were freshly cultivated from long-term storage on Sabouraud 4% Dextrose Agar plates (SDA; Sigma-Aldrich) at 23 °C and had been harvested in sterile Triton X-100 (Sigma-Aldrich, 0.05% in Milli-Q water). Spore germination was confirmed to be > 98% one day prior to each experiment.

## Pupal CHC profile

**Pupal infection.** To determine how the pupae's CHC profile changes with the presence or absence of workers in both castes, we individually exposed each pupa to the fungal pathogen by placing it on a glass slide, applying the fungal suspension and rolling the pupa in the suspension using soft forceps. Control pupae were treated the same, except for using Triton X-100 solution as a sham treatment. Pupae were then left to air-dry on the slide before putting them in a plastered dish (90 x 35 mm, Licefa GmbH) for three days before the start of the experiment. We used 1 μL of a 1 x $10^6$ spores/mL fungal suspension for the exposure of worker pupae and 2 μL of a 1 x $10^7$ spores/mL fungal suspension for queen pupae, and the same respective volumes of sham solution for the control pupae. Using the below-detailed fungal load determination by quantitative real-time PCR (qPCR), we determined that this exposure procedure resulted in an approximately 10-fold higher exposure dose for the queen pupae than the worker pupae (queen pupae: mean $1.587 \times 10^{-3}$ ng/μL, maximum $2.232 \times 10^{-3}$ ng/μL, $n = 3$; worker pupae: mean $1.603 \times 10^{-4}$ ng/μL, maximum $2.158 \times 10^{-4}$ ng/μL, $n = 7$). Our application dose of 1 μL of a 1 x $10^6$ spores/mL fungal suspension resulted in a realised exposure dose per worker pupa of approximately 95 spores[22], allowing us to infer an approximate load of 940 spores per queen pupa, both presumably being within natural exposure range, given single cadavers produce up to 12 million spores[59] and soil can contain 5000 spores / gram[60]. We exposed queen pupae to this higher spore load, since they are much larger than worker pupae (mean queen pupa length: 5.7 mm, width: 2.5 mm; $n = 4$; mean worker pupa length: 2.9 mm, width: 1.1 mm; $n = 4$, resulting in approximately 4-fold greater surface area and 9-fold higher volume of queen pupae compared to worker pupae). Preliminary data also showed that approximately half this dose was insufficient to induce a successful infection in queen pupae. This is in line with reports that, generally, social insect queens have higher immunocompetence than workers[34] and, in particular, that *Lasius* queens show higher immune gene expression than workers[33].

To allow for the successful establishment of infection, pupae were kept in plastered dishes (90x35 mm), after exposure, for three days in the absence of workers (as in ref. 22), as otherwise the sanitary care provided by accompanying workers would interfere with the infection process due to spore removal and disinfection[61,62]. Equally, control pupae were kept without workers for the same duration after their exposure to the sham treatment to ensure they were the same age as the fungus-treated pupae. This incubation period and all following experimental procedures took place in a temperature and humidity-controlled room at 23 °C, 65% relative humidity (RH) and a 12 h day/night light cycle and the plastered dishes were watered every 2-3 days to keep humidity high.

**Experimental setup.** After the three days of fungal (or sham) infection establishment, individual pupae were placed into a plastered dish (35x10mm; Falcon or SPL Life Sciences), either alone or with two workers that originated from the same colony as the pupa, and which had been fed with $^{13}$C glucose as detailed below. Using a full-factorial design, we thereby set up four treatment groups: infected pupae without workers (I + W-), infected pupae with workers (I + W + ), control pupae without workers (I-W-) and control pupae with workers (I-W + ). From each treatment, we sampled pupae over a 42-hour period after setting them up with the workers, at seven time points in intervals of six hours (i.e., at 6, 12, 18, 24, 30, 36 and 42 h). At each time point, we collected both infected and control pupae and recorded if a pupa had been unpacked in the presence of workers. This regular sampling was performed to ensure we only analysed pupae that were still able to modulate their cuticular hydrocarbons (CHCs), and had not yet been damaged by the destructive disinfection by the workers. To this end, we excluded worker pupae that appeared themselves damaged or dead, as identified by a 'shrivelled' morphology or a change in colouration ($n = 27$). We also excluded replicates, where one or both accompanying workers died during the experiment ($n = 18$), and three pupae were misplaced during the experiment. From these remaining 358 replicates, 35 had to be excluded after chemical analysis as they did not match our quality criteria as detailed below, leading to a final sample size of 323 worker pupae, of which 69 I + W-, 133 I + W + , 64 I-W-, 57 I-W + , from both the Jena ($n = 168$) and the Seva ($n = 155$) populations. For the queen pupae, the same quality criteria were applied during the experiment. Only one queen pupa did not match the chemical criteria, leading to 103 analysed queen pupae, of which 19 I + W-, 45 I + W + , 19 I-W-, 20 I-W + , all from the Jena population). For each pupa, we also obtained the pooled CHC profiles of its two accompanying workers (190 pools of the 380 accompanying the worker pupae and 65 pools of the 130 workers accompanying the queen pupae) to obtain their $^{13}$C integration levels. We froze each pupa with its cocoon (as in Pull et al. 2018[22]) in a brown glass vial (1.5 mL, Agilent) closed with one-component-closure-caps (Agilent) at − 70 °C, to store them for later chemical extraction for determination of the pupal surface chemicals, followed by DNA extraction and quantification of the fungal load. For replicates containing workers, both workers were pooled in a single glass vial for later chemical analysis (detailed below). We conducted the experiment separately for worker and queen pupae, with each experiment lasting five days, from the time of exposure, over the three days of isolation to the last sampling point, each carried out in a single block.

**$^{13}$C-enrichment of accompanying worker-CHCs.** In this study, we wanted to compare the pupal CHC production depending on both their infection status (infected I + vs uninfected I-) and the presence or absence of attending workers (W + vs W-). As it is known that colony members exchange CHCs to form a colony "gestalt" odour[24,25], we developed a method to exclusively quantify the pupa-produced subset of the CHCs found on the pupae, removing any possible worker-applied compounds, in order to accurately determine pupal signalling. To do this, we used stable isotope labelling[63,64] of the worker compounds by feeding workers glucose enriched with $^{13}$C, which is a stable natural isotope of $^{12}$C, and accounts for 1.1% of environmental carbon. By feeding workers the $^{13}$C-enriched glucose (99% $^{13}$C compared to the natural abundance of ~1.1%), $^{13}$C becomes incorporated into the workers' CHC compounds at elevated levels. When these $^{13}$C-enriched workers tend normal pupae (i.e., pupae from stock colonies that had not been reared on labelled $^{13}$C-glucose), they transfer their $^{13}$C to the pupae, altering the natural $^{12}$C to $^{13}$C ratio in the pupal extracts (see below). Quantification of the $^{12}$C to $^{13}$C ratio of each peak of (i) the tending workers (to determine the exact $^{13}$C integration level into the workers' CHCs for each peak and replicate) and (ii) the pupa kept with these workers, will therefore indicate which proportion of the CHC is

from non-enriched pupal origin and which from enriched worker origin. By using the formula detailed below to exclude the worker-applied proportion, we were able to calculate the quantity of each CHC peak produced exclusively by each pupa, as a function of its infection status and the social context (presence or absence of workers).

Worker ants were put into experimental containers (90x35 mm) in groups of 50 (worker pupae experiment: $n = 20$ groups; queen pupae experiment: $n = 11$ groups) and fed exclusively with $^{13}$C-enriched glucose solution (D-Glucose-$^{13}C_6$, 99 atom %, Eurisotop; 10% weight/volume in water) for three weeks prior to being placed with the pupae in the experiment. Throughout these three weeks, the pupae, later used in the experiment, remained in the stock colonies, and hence had not been able to incorporate labelled glucose as larvae. After these three weeks, two workers from the same feeding dish were put with an individual pupa originating from the same stock colony, as detailed above.

### Chemical analysis of pupae-derived CHCs
**Gas chromatography–mass spectrometry.** We used gas chromatography–mass spectrometry (GC–MS) to determine the composition of CHCs of worker and queen pupae, as well as their attending workers (where appropriate, i.e., in I + W+ and I-W + ). To this end, we extracted pupal CHCs from individual worker and queen pupae, as well as the worker CHCs from the pool of both workers per replicate, by adding 90 μL n-pentane solvent (Supelco) to the vials in which they had been collected. Vials were closed with aluminium-faced silicon septa caps (Agilent). Extractions were performed with the solvent for 5 min under gentle agitation at room temperature. The supernatant was transferred to glass vials with 350 μL glass inserts and sealed with aluminium-faced silicon septa (Agilent). The n-pentane solvent contained two internal standards at the beginning and the end of the known range of cuticular hydrocarbons of *L. neglectus* (C27 - C37)[22,26,27], n-tetracosane and n-hexatriacontane at 0.1 μg/mL concentration (both C/D/N Isotopes), both fully deuterated to enable spectral traceability and separation of internal standards from ant-derived substances. The extracts from the different treatment groups were run in a randomised manner, intermingled with blank runs (containing only n-pentane), and handling controls, using GC–MS (gas chromatograph GC7890 coupled to mass spectrometer MS5975C; Agilent).

A liner with one restriction ring filled with borosilicate wool (Joint Analytical Systems) was installed in the programmed temperature vaporisation (PTV) injection port of the GC, which was pre-cooled to − 20 °C and set to solvent vent mode, and 50 μL of the sample extracts were injected automatically into the PTV port at 40 mL/s using an autosampler (CTC Analytics, PAL COMBI-xt, CHRONOS 4.2 software; Axel Semrau) equipped with a 100 μL syringe. Directly after injection, the temperature of the PTV port was increased to 300 °C at 450 °C/min, thereby vaporising the sample analytes and transferring them to the column (DB-5ms UI; 30 m × 0.25 mm, 0.25 μm film thickness; Agilent) at a flow rate of 1 mL/min. Helium was used as the carrier gas at a constant flow rate of 3.0 mL/min for 2 min, then decreased to a constant flow rate of 1.1 mL/min at 100 mL/min[2]. The oven was programmed to hold 35 °C for 4.5 min, then ramp to 325 °C at 20 °C/min, and hold this temperature for 10 min. The GC–MS transfer line was set to 325 °C, and the mass spectrometer (MS) was operated in alternating TIC-SIM mode with a scan range of 35–600 amu in total ion current (TIC) mode, using electron ionisation mode (70 eV; ion source 230 °C; quadrupole 150 °C, with a detection threshold of 150). Disentanglement of worker-applied and pupa-produced compounds (using the below detailed quantification of $^{12}$C and $^{13}$C) was based on the SIM (selected ion monitoring) mode, where the ions 57.1, 58.1, 59.1, 60.1, and 61.1 were selected. In addition, a C7-C40 saturated alkane mixture (in steps of 1 at a concentration of 0.1 μg/mL for each alkane in n-pentane) was run as an external standard, to enable calculation of Kováts Retention Indices (RIs)[65,66] and to correct for possible shifts in retention time (RT).

Compound peaks were extracted from the total ion current chromatograms (TICCs) of representative sample runs, using a deconvolution algorithm (MassHunter Workstation, Qualitative Analysis B.07.00; Agilent). CHC identification was performed by comparing the mass spectrum and RI of each compound by manual interpretation of diagnostic ions, as well as by comparison to previous cuticular hydrocarbon (CHC) compounds identified for *L. neglectus* adult workers[26,27] and worker pupae[22], as detailed in Supplementary Table S13 and with representative mass spectra deposited under doi.org/10.15479/AT-ISTA-20471. For reasons outlined below, this study includes only a subset of previously described CHCs, specifically those that met our criteria for accurately determining $^{13}$C levels using the method we developed to disentangle worker-applied from pupa-produced CHCs.

Besides the pupal CHCs, we also analysed the CHC peaks of their two accompanying workers by the above-mentioned GC–MS procedures, using automatised integration of the peak area of each peak and standardisation of the amount of each peak to its closest internal standard eluting before (MassHunter Workstation), thereby correcting for possible injection differences. For each peak, we quantified the $^{12}$C and $^{13}$C proportion based on the SIM scans, using the generally highly abundant ion 57.1 as quantifier ion for $^{12}$C (as it represents the fragment $^{12}C_4H_9$), and the sum of ions 58.1, 59.1, 60.1 and 61.1 as quantifier ions for $^{13}$C, as they reflect the level of $^{13}$C enrichment ($^{13}C_1{}^{12}C_3H_9$, $^{13}C_2{}^{12}C_2H_9$, $^{13}C_3{}^{12}C_1H_9$ and $^{13}C_4H_9$ respectively; simplified scheme due to low natural abundance of other possible isotopes). Since $^{13}$C makes up 1.1% of environmental carbon, its natural level in $C_4H_9$ – and hence the baseline found in non-enriched samples (Supplementary Fig. S1) – is 4.4%.

Using this method, we found that the CHC profiles of the workers fed with $^{13}$C-labelled glucose do indeed show peak-specific $^{13}$C levels increased above the natural values of ~ 4.4%. Moreover, these workers transferred CHCs to the pupae during social contact: pupae kept with workers showed increased $^{13}$C levels ( > 4.4%), whereas pupae kept without workers did not, which was equally true for worker (Supplementary Fig. S1a) and queen (Supplementary Fig. S1b) pupae. Notably, the intensity of $^{13}$C-enrichment differed between worker groups (likely depending on differences in feeding activity), yet the difference between individual workers within feeding groups was negligible (as workers of the same feeding group differed only by a mean of 2.2% in their proportion of $^{13}$C in peak C33:2 and 3.1% in C33:1, as determined by quantification of $n = 19$ additional pairs of workers from 4 feeding groups). To determine the level of $^{13}$C incorporation in the two workers per replicate, we pooled both individuals for a joint extraction. This allowed us to obtain the replicate-specific worker $^{12}$C and $^{13}$C proportion for each of their CHC peaks, which was necessary to estimate the amount of each peak produced by each pupa.

### Subtraction of worker-transferred CHCs from total pupal profiles.
$^{13}$C labelling of worker-derived CHCs allowed us to subtract the transferred worker CHCs from the total peak amount measured in the pupae (pupa- and worker-derived). This enabled us to quantify the CHCs produced exclusively by the pupae, which we then compared between treatments. As only the workers but not the pupae had been $^{13}$C-enriched at the start of the experiment, the proportion of $^{13}$C in solely pupa-produced CHCs reflected the natural 4.4% in the four C atoms (as also confirmed by pupal CHCs in the absence of workers, Supplementary Fig. S1). Consequently, any $^{13}$C-content exceeding the natural background level of ~ 4.4% in a peak indicates application of the $^{13}$C-enriched CHCs from the labelled workers. Since we measured, for each peak and replicate, the level of $^{13}$C incorporation into the workers' CHCs resulting from their feeding of the $^{13}$C-labelled glucose, we could determine both the $^{13}$C proportion in workers (as quantified from the experiment) and in pupae (natural baseline of 4.4%). Given the measured value for total (pupa- and worker-derived) $^{12}$C and $^{13}$C per peak, we could therefore disentangle the pupa- and worker-derived contribution and reach the purely pupa-produced quantity ($A_{pupa}$) for each

peak by the following formula:

$$A_{pupa} = \frac{\varphi_{13}^W A_{12} - \varphi_{12}^W A_{13}}{\varphi_{12}^P \varphi_{13}^W - \varphi_{13}^P \varphi_{12}^W} \qquad (1)$$

where, for each peak:

$\varphi_{13}^W$ or $\varphi_{12}^W$ = the proportion of $^{13}C$ or $^{12}C$ of the peak measured in the accompanying workers (quantified from the replicate-specific worker extract)

$\varphi_{13}^P$ or $\varphi_{12}^P$ = the natural proportion of $^{13}C$ or $^{12}C$ in the unenriched pupal CHC (set to 0.044 and 0.956, respectively)

$A_{12}$ = the total amount of $^{12}C$ of the peak quantified from the pupal extract (reflecting both pupa- and worker-derived $^{12}C$)

$A_{13}$ = the total amount of $^{13}C$ of the peak quantified from the pupal extract (reflecting both pupa- and worker-derived $^{13}C$)

Note that this calculation was performed for each peak and replicate, using the amounts obtained by standardisation to the closest internal standard eluting before the peak (as detailed above). From the previously described CHCs of *L. neglectus* worker pupae[22], six compounds (3-methylhentriacontane; 13,23-dimethylpentatriacontane; 11,25-dimethylpentatriacontane; 7,11,23-trimethylpentatriacontane; *n*-hexatriacontane and one unidentified) could not be included in the current analysis, as they were either co-eluting with other compounds impacting quantification or $^{13}C$ was undetectable in many of the pupal and worker samples (note that four of these excluded compounds were analysable in the queen pupae due to higher compound abundance, yet their inclusion did not alter our findings so for consistency reasons we report only the CHCs that are conclusive for both worker and queen pupae). Worker and queen pupae shared the same peaks (Supplementary Table S13).

Furthermore, we could not include 35 worker and one queen pupae samples, which had been originally sampled, but for which an accurate assessment of their pupa-produced CHC quantity was not possible for all peaks (these pupae are already deducted from the final sample sizes reported above). This was either because – likely due to natural variation or measurement inaccuracies at low peak abundances – the $^{13}C$ proportion of some of their peaks had a higher value than that of accompanying workers (worker pupae: $n = 21$; queen pupae: $n = 0$), or as their $^{13}C$ was below detection threshold in some peaks, which were typically overall of low quantity (worker pupae: $n = 14$; queen pupae: $n = 1$). When the measured $^{13}C$ proportion of a peak was below 4.4% (similarly likely due to natural variation around the 4.4% average or measurement inaccuracies), which occurred in 10.68% of the worker pupae peaks (621 out of 5814) and 18.39% of the queen pupae peaks (341 out of 1854), we used the measured value as pupa-produced amount because workers cannot have transferred said peak. Similarly, when a peak was undetectable in both $^{12}C$ and $^{13}C$ forms, or only detectable in $^{13}C$, in the accompanying workers (134 cases in the worker pupae and 85 cases in the queen pupae), or when no workers were present in the treatment group (i.e., in all I + W- and I-W- groups), the total measured value in the pupa was taken as the pupa-produced amount. For each pupa-produced peak, we then calculated its relative proportion in the overall pupal chemical bouquet and report these relative abundances (standardised amount of the respective peak / sum of the standardised amounts of all 18 peaks). Our main focus was to see whether the relative abundances of the four CHCs that were identified by Pull et al. 2018[22] as possible candidate compounds for either a cue or signal were differentially modulated by the pupae in response to treatment.

**Immune gene expression**

We repeated the experiment for both worker and queen pupae, to analyse how the expression of three candidate immune genes depended on pupal caste, infection status and worker presence. Since we were not aiming at measuring pupal CHCs in this experiment, no

$^{13}C$-labelling of the accompanying workers was required. We used one gene representing each of the three stages of the immune reaction: β-1,3-glucan binding protein (*BGBP*) for pathogen recognition[29], peptidoglycan recognition protein SC2 (*PGRP-SC2*) for immune regulation[30,31] and the antimicrobial peptide Defensin 1 (*Def1*) for immune effector activity[32,67]. Several pupae were excluded from the analysis due to death or mould during the experiment ($n = 12$ queen pupae), incorrect developmental stage ($n = 3$ queen pupae), or poor RNA quality ($n = 9$ worker pupae). This resulted in a final sample size of 117 worker pupae (19 I + W-, 58 I + W + , 20 I-W-, 20 I-W + ) and 62 queen pupae (17 I + W-, 20 I + W + , 11 I-W-, 14 I-W+). All pupae and their corresponding workers originated from the Jena population.

**RNA extraction.** Before extracting their RNA, pupae were homogenised using a TissueLyser II (Qiagen) with a mixture of one 2.8 mm ceramic (VWR), and five 1 mm zirconia (BioSpec Products) and approx. 100 mg glass beads (425–600 μm, Sigma-Aldrich). Homogenisation was carried out in two steps (2 x 2 min at 30 Hz). Pupal RNA was extracted using either the Maxwell RSC simplyRNA tissue kit (Promega) or the AllPrep DNA/RNA Mini kit (Qiagen). Both kits were used following the manufacturer's instructions, using a final elution volume of 50 μL (Promega kit) or 14 μL (Qiagen kit) RNase-free water. To ensure removal of residual DNA contaminations we performed a DNase I treatment (Sigma-Aldrich). cDNA was synthesised using iScript cDNA synthesis kit (Bio-Rad) according to the manufacturer's recommendations, using 11 μL of DNase I treated RNA as starting material.

**Gene expression.** Expression levels of the three immune genes *BGBP*, *PGRP-SC2* and *Def1*, as well as the housekeeping gene 28S Ribosomal Protein S18a (*RP-S18a*[68]), were analysed in 20 μL reaction volumes using KAPA SYBR Fast qPCR master mix (Roche) and 4 pmol each of specific primers (Sigma-Aldrich; for primer sequences see Supplementary Table S14) on a Bio-Rad CFX96 real-time PCR detection system. Two microliters of the either undiluted or 1:10 diluted cDNA (4 μL for some samples, as expression values for defensin were low) were added per reaction, and each sample was analysed in duplicate or triplicate wells. Each run contained an absolute negative control. Primer efficiency was determined to lie between 90 and 105% for all primer sets using standard curves of 10-fold dilutions. The following amplification programme was used: 95 °C for 5 min, followed by 40 cycles of 10 s at 95 °C and 30 s at 55 °C (*Def1*: 20 s at 60 °C). Following each run a melting curve analysis was run to monitor primer specificity. Immune gene expression levels were determined using the standard curve method and were normalised to the expression level of the housekeeping gene *RP-S18a*.

**Pathogen load quantification**

We further quantified the 'fungal load' of all fungus-treated worker and queen pupae from the CHC experiment after having extracted their surface chemicals (as detailed above). One worker pupa was lost during DNA extraction, leading to a final sample size of $n = 201$ worker and $n = 64$ queen pupae. We also analysed all sham-treated pupae (except for three samples that could not be analysed due to technical issues; leading to 119 control worker pupae and 38 control queen pupae), and used their highest value to set the concentration threshold above which we could confidently identify fungal infection and perform an accurate fungal load quantification using real-time PCR (qPCR), quantifying absolute fungal DNA load using the standard curve method (detailed below). With a value of $7.2 \times 10^{-4}$ ng/μL this threshold value was approximately 500-fold and 3000-fold lower than the mean fungal load of the worker pupae ($3.9 \times 10^{-1}$ ng/μL) and queen pupae (2.3 ng/μL), respectively. We also quantified the fungal load of seven worker and three queen pupae samples frozen straight after exposure, to obtain the worker pupae and queen pupae-specific 'exposure dose' (reported above). As these exposure doses differed between the

castes, we calculated the 'infection load' for each pupa as its measured fungal load relative to the maximum caste-specific exposure dose to determine the infection progression after exposure.

**DNA extraction and qPCR.** We extracted and quantified *Metarhizium* DNA per infected ant pupae by targeting the sequence of the fungal ITS2 gene as in Giehr et al. 2017[69]. Prior to DNA extraction, the samples were homogenised as detailed above. DNA extractions were performed using the DNeasy 96 Blood & Tissue Kit (Qiagen), following the manufacturer's instructions, with a final elution volume of 50 µl Buffer-AE. qPCR was performed using primers targeting the *Metarhizium brunneum* ITS2 rRNA gene region as in Giehr et al. 2017[69] (primer sequences: Supplementary Table S14). Reactions were performed in 20 µL volumes including 1x KAPA SYBR® FAST qPCR Master Mix (Roche), 3 pmol of each primer (Sigma-Aldrich) and 2 µL of extracted DNA. The amplification programme was initiated with a first step at 95 °C for 5 min, followed by 40 cycles of 10 s at 95 °C and 30 s at 64 °C. Quantification was done based on the standard curve method, using standards covering a range from $10^{-1}$ to $10^{-5}$ ng/µL fungal DNA (measured using a NanoDrop spectrophotometer, Thermo Fisher Scientific). Each run included the standards and a negative control. Samples, as well as standards and controls, were run in triplicates. Specificity was confirmed by performing a melting curve analysis after each run.

### Generation of signal and non-signal extracts

To test for a causal relationship between the CHC profile of the pupae and worker unpacking, we obtained the chemical extract of worker pupae that had been infected and unpacked by the workers ('signal extract') and of sham-treated pupae that did not elicit worker unpacking ('non-signal extract'). We transferred the extracts onto worker and queen pupae in a bioassay, testing if the CHCs by themselves elicited unpacking behaviour towards the otherwise healthy brood. We also characterised the isomeric compositions of the identified signal peaks C33:2 and C33:1, and determined how they differed in the signal and non-signal extract (as detailed below).

**Generation of signalling and non-signalling pupae.** The bioassay, in which we experimentally transferred the extracts onto worker and queen pupae (as detailed below), required a high number of signalling worker pupae. To generate these, age-controlled worker pupae (as described above) were exposed to 1 µL of a spore suspension at a concentration of $1 \times 10^9$ spores/mL. This corresponds to the 'high dose' defined by Pull et al. 2018[22], in contrast to the 'medium dose' $(1 \times 10^6)$[22] used in the experiments above. This was appropriate since Pull et al. had established that the high dose results in a higher unpacking rate, while worker behaviour remains otherwise unchanged[22]. In addition, our detailed chemical comparison during method establishment found no distinguishable difference in CHC peak patterns between doses, indicating equivalent CHC compositions. Due to this higher exposure dose and hence faster disease development, we also shortened the isolation period of the pupae to two days before adding two nestmate workers. Sham-treated pupae were produced as in previous experiments, and workers were added to create the I-W+ treatment as the closest control of non-signalling pupae to the signalling group (I + W + ). All fungal and sham-exposed pupae were individually placed in plastered dishes (35 x 10 mm, SPL Life Sciences) directly after the applied treatment had dried. After worker addition, we checked pupae for unpacking regularly every 6-10 h over a period of 42 h, to make sure signalling pupae were collected shortly after unpacking and before being destroyed by the workers. We collected 'confirmed signalling pupae' that had been unpacked after fungal treatment and 'confirmed non-signalling pupae' that remained cocooned after sham-treatment. Pupae were collected in clear glass headspace vials (10 mL, Thermo Fisher Scientific) and frozen at − 70 °C for further use in the bioassay and chemical analysis.

### Bioassay determining unpacking of healthy pupae after extract transfer

**Generation of CHC extract for transfer onto pupae.** For application onto healthy worker and queen pupae, we extracted a total of 681 worker pupae (343 signalling and 338 non-signalling) in 22 pools of on average 31 pupae (min. 25, max. 37), always containing only pupae from the same treatment and population (120 pupae from Seva and 561 from Jena). The pools were extracted as detailed below, and used to transfer signal extract to healthy pupae that had been randomly sampled from their colonies the day before the bioassay, age-controlled as above, and kept overnight with workers from the same colony in plastic containers (90 x 35 mm) with humidified, plastered ground. In total, the signal extract was transferred to 113 pupae and the non-signal extract to 102 pupae, so that each treated pupae received an extract equivalent of approximately three source pupae (as detailed below).

Extracts were generated by adding 300 µL *n*-pentane per pupa to the 10 mL clear glass vials containing the pooled pupae (average *n* = 31 / vial, as detailed above), and incubated at room temperature for 10 min, after which the supernatant containing the extracted CHCs was transferred to a brown glass headspace vial (10 mL, Thermo Fisher Scientific). We then rinsed the original extraction vial to dissolve all remaining CHCs with another 1 mL of *n*-pentane and added it to the vial containing the supernatant, before evaporating the extract in a dry bath (Labnet International) at 45 °C, partially aided by blow-drying with nitrogen gas (taking approx. 1 h). We dissolved this dried extract by adding 800 µL *n*-pentane and vortexing at maximum speed for approx. 15 sec and transferred it to a 1.5 mL brown glass vial. To dissolve all CHCs, we washed the original vial a second time with 800 µL *n*-pentane, adding the 800 µL into the vial. After solvent evaporation, the residue was taken up in two washes of 80 µL *n*-pentane and transferred to a new 1.5 mL glass vial, now containing a 150 µL glass microinsert (Chroma Globe). After solvent evaporation, the walls of the microinserts were rinsed with 30 µL *n*-pentane and dried one last time, thereby concentrating the CHCs at the bottom of the insert. The dry extracts were stored at − 20 °C under nitrogen gas until use in the bioassay the day after.

**Extract transfer onto healthy pupae.** We transferred the signal extract to 78 worker pupae (with a final sample of 76 since two got damaged during the procedure) and 35 queen pupae, and non-signal extract to 64 worker and 38 queen pupae, to test if the CHC profiles per se contained the unpacking trigger and if unpacking could also be elicited in the queen pupae.

For application onto the healthy pupae, we dissolved the extracts by adding 1 µL *n*-pentane per extracted pupa, leading to an average volume of 31 µL (min. 25 µL, max 37 µL) per microinsert, depending on initial pool size. Previous method establishment had revealed that this volume in the microinsert minimises evaporation of the volatile *n*-pentane. Therefore, 8 – 13 healthy pupae could be treated per batch by four consecutive applications of a 0.5 µL droplet of either signal or non-signal extract, leading to a total application volume of 2 µL per pupa, with the same volume applied to queen and worker pupae. We confirmed that the order of application of the extract had no effect on a pupa's unpacking probability, revealing that the variation inherent to the application procedure did not lead to a biological effect. We applied the final volume of 2 µL in four steps of 0.5 µL each to ensure that the applied extract dried on the pupal surface and did not run off to the underlying glass plate (cleaned with *n*-pentane between batches), on which all receiving pupae of each batch were placed before the transfer. To achieve this high yield, we cooled the extract vials before use at 4 °C and kept them in a cool metal block on ice during the experiment, as well as used a pre-cooled low-volume volume-syringe (Trajan Scientific), stored at − 20 °C before each round of application. After application, all treated pupae per batch were transferred

individually to a dish (Ø = 35 mm) with plastered ground, and two workers from the same colony were added to each pupa. We ran the experiment separately for the worker and the queen pupae, each in multiple blocks (three blocks for the worker pupae and four for the queen pupae). After blinding the treatment information and intermingling the batches, we checked each pupa for unpacking by the two workers within 12 h after application for the worker pupae and 4 h for the queen pupae, as method establishment revealed that the ants reacted about 3 times faster to the manipulation of queen than worker pupae. In fact, after 4 h, only 3.9% of signal-treated and 1.5% of the non-signal-treated worker pupae had become unpacked, whereas in the same time, 14.3 % of the signal-treated and 2.6% of the non-signal-treated queen pupae were unpacked.

**Comparative treatments.** In addition to the transfer of worker pupae-produced extracts, we performed the following applications, all on worker pupae from Jena (total $n = 62$). Application of pure $n$-pentane as a negative control did not induce unpacking in any of the pupae ($n = 21$), confirming that the procedure by itself does not induce unpacking. We also applied the compound (Z)-tritriacont-10-ene ((Z)-10-C33:1; LGC Group), which elicits removal of treated honeybee pupae from their brood cell[40], but which we did not find increased in our signal extract (see below and Supplementary Table S12). We applied (Z)-10-C33:1, dissolved in $n$-pentane at either a concentration of 2.5 ng/µL ($n = 21$) or 4 ng/µL ($n = 20$), and thus in a final dose of 5 resp. 8 ng per pupa, to remain within the natural abundance of C33:1 (RI 3281) in our samples of approx. 9 ng/pupa.

## Isomer characterisation of the CHC signal
We used one pool each of 20 signalling resp. non-signalling pupae (from Jena) to analyse the isomeric composition of the signal CHCs.

**Generation of extract for isomer characterisation.** We obtained signal and non-signal extract for a pool of 20 worker pupae each, by adding 300 µL $n$-pentane per pupa to the 10 mL clear glass vials, and incubation at room temperature for 10 min, after which the supernatant containing the extracted CHCs was transferred to a brown glass headspace vial (10 mL). After solvent evaporation, the samples were redissolved in 100 µL $n$-pentane. To remove polar compounds, we then passed these extracts through a conditioned SPE (solid-phase extraction)-column filled with unmodified silica gel (CHROMABOND® SiOH, 1 mL, 100 mg, Macherey-Nagel). The hydrocarbons were then eluted with 4 mL of $n$-pentane, the solvent evaporated, and the samples redissolved in 600 µL $n$-pentane (30 µL per pupal equivalent), to be used for the generation of native chromatograms and further DMDS-derivatisation.

**GC–MS of native (non-derivatised) signal and non-signal extract.** From each sample, 60 µL were transferred to an autosampler vial (with fixed insert 350 – 10 µL, clear glass, Agilent). The solvent was evaporated and the samples redissolved in 110 µL $n$-pentane containing our deuterated internal standards. The samples were run on the GC–MS, with the following changes to the above method: the PTV injection port was set to 34 °C and the alternating TIC-SIM mode was used, with selected ions being 66.2, 460.9 and 462.9, with a dwell time of 20 ms each. In the same sequence and with the same settings, we ran the C7-C40 saturated alkane mixture as above, yet also (Z)-tritriacont-10-ene ((Z)-10-C33:1, 1 µg/mL in $n$-pentane), which had been identified as a compound eliciting hygienic behaviour in honeybees[40]. Overlay of the SIM chromatograms of these native signal and non-signal extracts for C33:2 ($m/z$ 460.9; Fig. 2c) and C33:1 ($m/z$ 462.9, Fig. 2d) was used to compare peak shapes, indicative for differences in both quantitative abundances and isomeric compositions between the two extracts. Overlay of the signal extract with the (Z)-10-C33:1 standard (Supplementary Fig. S6) was used to indicate the retention time of the C33:1

isomer with double bond at position 10, (i) allowing interpretation of the relative positioning of the other C33:1 isomers (ranging from double bonds at positions 13 to 7) in our samples, and (ii) revealing that also the C33:1 isomers in our samples similarly are present in (Z)-configuration. The relative retention times of C33:2 vs C33:1 in our samples and the fact that the (Z)-form is also the only so far reported isomer configuration found for alkenes and alkadienes across the social insects[46], further makes the (Z-Z)-configuration highly likely also for our C33:2 isomers.

**GC–MS of DMDS-derivatised signal and non-signal extract.** To identify the isomers of C33:2 and C33:1 in our samples, we subjected 420 µL of the extracts, concentrated to 50 µL, to DMDS derivatisation. To obtain the positions of the double bonds of the isomers, we followed the protocol by Carlson et al. 1989[70]. The 50 µL of concentrated samples in $n$-pentane were mixed with 50 µL DMDS (Sigma-Aldrich) and 25 µL of a 6% iodine solution (w/v iodine in diethyl ether, both Sigma-Aldrich). The reaction mixtures were held in 1.5 mL vials closed with PTFE/silicone screw caps (Agilent) at 40 °C for 5 h, before they were diluted with 1 mL $n$-pentane and transferred to a 10 mL clear glass headspace vial (Thermo Fisher Scientific), where they were further diluted with $n$-pentane to a total volume of 4 mL. The same volume of 5% sodium thiosulfate pentahydrate (w/v in water, both Sigma-Aldrich) was added, the mixture vortexed, the organic phases removed and dried over anhydrous sodium sulfate (Sigma-Aldrich). The extracts were decanted, the solvent evaporated, and the samples redissolved in 110 µL $n$-pentane for GC–MS measurement. For this, the GC–MS method was adapted as follows: the PTV injection port was set to 34 °C and the final temperature was set to 330 °C. The oven programme was ramped to 350 °C (20 °C/min) and held on this temperature for 15 min. The GC–MS transfer line was set to 355 °C and the scan range of the MS was extended to 700 amu. The selected ions for SIM mode were 61.0, 145.1, 173.1, 187.1, 201.1, 215.2, 229.2, 507.6, 509.6, 556.7 and 648.7, with a dwell time of 12 ms each. The C7-C40 saturated alkane mixture was used to estimate the Kováts RIs for the DMDS-adducts of the X-C33-monoenes. Since the X,Y-C33-dienes eluted after the C40 reference alkane, an additional alkane (C44, contained in a C12-C60 mixture (ASTM 5442 C12-C60 standard, Supelco) was measured to extend the reference range for RI estimations. For the di-adducts eluting beyond C44, retention times of hypothetical $n$-alkanes were extrapolated using a polynomial regression model fitted to the retention times of the known $n$-alkanes. The extrapolated retention times for C47 and C48 were then used in the Kováts equation to calculate the RIs of the C33:2-DMDS-di-adducts. While no previous isomeric information was available for C33:2 of *L. neglectus*, we found all C33:1 isomers previously described for *L. neglectus* workers[26], plus tritriacont-9-ene, 9-C33:1) in our pupal samples. We show the structures of the annotated C33:2 isomers and C33:1 isomers in the inferred most likely (Z-Z)- and (Z)-configuration (Supplementary Fig. S4a for C33:2 and S4b for C33:1, respectively) and the mass spectra for the two dominant C33:2-DMDS-di-adducts (Supplementary Fig. S5a) and for all annotated C33:1-DMDS-mono-adducts (Supplementary Fig. S5b).

**Annotation of X-Y-C33:2 and X-C33:1 isomers.** We followed the method by Carlson et al. 1989[70] for isomer annotation. SIM was chosen to extract the molecular ion $(M)^+ = m/z$ 648.7, as well as the diagnostic fragment ion $(M-141)^+ = m/z$ 507.6 common to all DMDS-di-adducts of X,Y-C33:2. The corresponding $(A)^+$ and $(D)^+$ fragment ions were determined by inspection of their SIM chromatograms and extracted ion chromatograms (EICs). When the supposed fragment ion pair showed comparable retention behaviour (retention time, peak shapes, parallel increase and decrease of their abundances over the peak width) and also plausible abundance ratios of the $(A)^+$ and $(D)^+$ ions, they were considered as a matching ion pair, indicating the double bond positions in the respective C33:2 isomer. SIM was further chosen

to extract the molecular ion $(M)^+ = m/z$ 556.6, as well as the diagnostic fragment ion $(M-47)^+ = m/z$ 509.6 common to all DMDS-mono-adducts of X-C33:1. The $(A)^+$ and $(B)^+$ fragments for the mono-adduct of each positional isomer were determined by inspection of the averaged mass spectrum and the respective extracted ion chromatograms (EICs). When the supposed $(A)^+$ and $(B)^+$ fragments ions showed the same retention behaviour (as above for C33:2) and fitted mathematically $((A)^+ + (B)^+ = (M)^+)$, we considered them as a matching ion pair, indicating the double bond position in the respective C33:1 isomer.

**Determination of the signal peak isomers.** Whereas all C33:2 co-elute to form a single peak, tritriacontene (C33:1) separates into two peaks with identical mass spectra in the GC-MS chromatogram, of which only the first (RI 3281, Supplementary Table S13) was found to be significantly upregulated in unpacked pupae by Pull et al. 2018[22]. DMDS-derivatisation (Supplementary Table S12) and retention time comparison to the synthetic (Z)-10-C33:1 in the native chromatograms (Supplementary Fig. S6) revealed that this early C33:1 peak likely represents a mixture of isomers ranging from 13- to 9-C33:1. In contrast, the second, unaltered peak (RI 3289) is most plausibly attributed to the 7-C33:1 isomer. This assignment is supported by three lines of evidence: (i) in the derivatised data, the six identified isomers (Supplementary Fig. S4) eluted in order from 13 to 7, with the 7-isomer eluting distinctly after the 13- to 9-isomer cluster; (ii) overlay of the synthetic (Z)-10-C33:1 with the native extract revealed a shoulder indicating a residual compound trailing beyond the (Z)-10-C33:1 peak (Supplementary Fig. S6), suggesting that the first isomer cluster ends with the 9-isomer; and (iii) the relative abundance of the 7-isomer in the derivatised data (23% in non-signalling pupae and 18% in signalling pupae) closely matches the abundance of the second C33:1 peak in the non-derivatised data (21% in non-signalling and 16% in signalling pupae). The isomer ratio changes from the non-signalling to the signalling state (Supplementary Table S12) were therefore calculated based only on the 13- to 9-C33:1 isomers, as the 7-isomer is most likely not part of the biologically relevant pupal signal.

**Abundance ratios of isomers in the signal peaks.** The EIC- and SIM-abundances of the diagnostic ions $(A)^+$ and $(B)^+$ for DMDS-mono-adducts of the X-C33:1 isomers and $(A)^+$ and $(D)^+$ for DMDS-di-adducts of X,Y-C33:2 isomers were determined automatically (MassHunter), manually verified, and the integrations corrected, where necessary. We normalised the EIC abundance data for the $(A)^+$ fragment of the eight C33:2-DMDS-di-adducts, as well as for the five C33:1-DMDS-mono-adducts of the 13- to 9-position isomers that we found to underlie the first C33:1 peak of average RI 3281 in the non-derivatised chromatogram, to the total sum of their respective $(A)^+$ fragment abundances per extract. Specifically, for each adduct, its relative abundance was calculated by dividing its individual $(A)^+$ fragment abundance by the sum of the $(A)^+$ fragment abundances of either all C33:2-DMDS-di-adducts or the five C33:1-DMDS-mono-adducts of the RI 3281 cluster in its respective extract, yielding a normalised value between 0 and 1 for each adduct per extract, allowing us to assess the relative distribution of the C33:2 and C33:1 isomers for of the two signal peaks (C33:2 and C33:1 RI 3281) in both the signal and non-signal extract. Subsequently, these normalised isomer abundances in the signal extract were divided by the corresponding normalised abundances in the non-signal extract, receiving the isomer-specific ratios showing how strongly the isomer was up- resp. downregulated in signalling pupae (Supplementary Tables S11, S12).

Ordering details for used consumables can be found in Table S15.

**Statistical data analysis**
Statistical analyses were performed using R version 4.3.2 and R Studio version 2023.12.1.402. For all models, we checked the necessary assumptions by viewing histograms of data, plotting the distribution of model residuals, checking for unequal variances and the presence of multicollinearity and assessing models for influential observations. Where applicable, we transformed the data to obtain normality, using the package 'bestNormalize' (vs 1.9.1)[71] (using either order-norm, standardised box-cox, log, square-root, Yeo-Johnson, or arcsinh transformations).

For all models, we first tested the significance of the overall model compared to a null model (only including the intercept and variables for which we wanted to control) to test whether or not main effects or their interaction overall had a significant effect. For sampling time, we achieved balanced groups and higher statistical power by combining the timepoints 6-12 h to an early, 18–24 h to a middle and 30–42 h to a late 'infection period' in all analyses, except the time-resolved Cox Proportional Hazard Model. When both populations (Jena, Seva) were used in the experiment, we accounted for a population effect by including population as a control factor in the null model, since inclusion as a random effect would require at least five levels[72]. This was not required for the queen pupae, as they were only available from the Jena population. When multiple models were performed on related data, such as one model each for the four candidate peaks for both worker and queen pupae separately (Supplementary Tables S1a, b; S4a, b), or for the 14 non-candidate peaks of the queen pupae (Supplementary Table S6), we corrected at the model level (full-null model comparison) for multiple testing using the Benjamini-Hochberg procedure[73] to protect against a false discovery rate (FDR) of 5%. When the overall model remained significant after correction for multiple testing, we proceeded to test the significance of interaction and, where applicable, the main effects, using likelihood ratio (LR) tests[74] using the 'lmtest' package (vs 0.9.40)[75]. For models with significant interaction, the main effects cannot be reliably interpreted and are hence not reported. Within models, $p$-values of the interaction or main effects remained uncorrected, yet all post hoc comparisons were again corrected for multiple testing. Effect sizes were only calculated when a significant effect was found, using the packages 'rstatix' (vs 0.7.2)[76] and 'emmeans' (vs 1.5.4)[77]. Unless otherwise stated, all reported $p$-values are two-sided, corrected for multiple testing where applicable and reported as exact values. Figures were created using R version 4.5.0 and R Studio version 2023.06.0.421 with packages 'ggplot2' (vs 3.5.2)[78], 'dplyr' (vs 1.1.4)[79], 'ggpubr' (0.6.0)[80] and 'Rmisc' (vs 1.5.1)[81].

**Chemical profile.** To determine whether the pupae-produced amounts of the four candidate CHCs identified by Pull et al. 2018[22] differed depending on infection status and presence/absence of workers – in particular, if the CHC abundances were different in infected pupae in the presence (I + W + ) or absence (I + W-) of workers, and compared to the noninfected controls (I-W + , I-W-) – we ran a linear model per peak, testing for a significant interaction between our two main effects, thereby controlling for population (Jena, Seva) for the worker pupae and infection period (early, middle, late) for both worker and queen pupae by inclusion into the null model. Since we found no evidence of signalling in the four candidate peaks previously identified for worker pupae by Pull et al.[22] in the queen pupae (Supplementary Tables S4a, S5), we expanded our analysis in queen pupae to include the remaining 14 peaks (Supplementary Table S6).

Prior to statistical analysis, we obtained normality in the worker pupae by transforming all four candidate peaks by the ordered quantile normalisation. For the queen pupae, we had to use different transformations for different CHCs, i.e., box-cox transformation for C33:2, C33:1 (RI 3289), and 13MeC33, log transformation for 3MeC29, C30, C31, C33:1 (RI 3281) and 3MeC33, square root transformation for C35:2, ordered quantile normalisation for C27, C28, C29, C33 + 13MeC33:1, C34 + 14MeC34:1 + 12MeC34:1, C35:2 + C35:1, C35 + 13MeC35:1, C37 + 13MeC37:1 and arcsinh transformation for 3MeC33:1.

As we found that signalling was restricted to the two immune-associated peaks (C33:2 and C33:1) in the worker pupae, we wanted to test whether the combined differences of these two signalling peaks would lead to a significantly different chemical profile of the infected

pupae in worker presence (I + W + ) to the other groups. To this end, we carried out a combined analysis aggregating C33:2 and C33:1 into one analysis after having z-transformed each compound separately. To obtain normality, worker pupae data were then transformed using the ordered quantile normalisation, and queen pupae data using the Yeo-Johnson transformation. We ran linear mixed models (LMM) using the 'lme4' package (vs 1.1.36)[82] with the same variables as above, yet in addition accounting for peak (C33:2, C33:1) in the null model. We further included sample ID as a random effect to control for the non-independence of data points. After finding a significant interaction in the worker pupae (but no significance in the queen pupae, Supplementary Table S5), we carried out all-pairwise post hoc tests for the worker pupae (Supplementary Table S2) using the 'emmeans' package[77].

To determine if individual pupal infection load affected the pupal amount of the signalling peaks C33:2 and C33:1 in the worker pupae, we ran separate linear models (LM) for each peak, controlling for population, worker presence and infection period in the null model for the subset of the infected (I + ) worker pupae (Supplementary Table S1b). We did the same analysis for the infected (I + ) queen pupae for consistency, except here not having to control for population (Supplementary Table S4b).

**Unpacking of pupae.** Cox proportional-hazards regression analysis by use of the package 'survival' (vs 3.5.7)[83] was used to determine, for both the worker and the queen pupae, whether unpacking occurred differently for the infected than the control pupae, again accounting for the ant population (Jena, Seva) in the worker pupae. We calculated the hazard ratio of how much more frequently unpacking occurred in the infected vs control pupae only for the worker pupae, as we found no significant difference in the queen pupae.

**Immune gene expression.** To test for the effect of infection and worker presence on pupal immune gene expression, we conducted separate linear models (LMs) for both castes and each immune gene (*BGBP*, *PGRP-SC2*, and *Def1*), after having achieved normality by ordered quantile normalisation. Since we found no significant interaction between infection and worker presence in either caste for any gene (Supplementary Tables S3, S8), we proceeded to test the significance of the two main effects. In all models, we controlled for the infection period, whereas we did not have to account for population, as only one population (Jena) was used. As the three genes did not differ in their expression pattern (worker pupae: Supplementary Table S3a, queen pupae: Supplementary Table S8a), we calculated the combined immunity value per pupa by averaging the z-transformed normalised expression values of the three genes. We then reran the above models for each caste (Supplementary Tables S3b, S8b).

To test for a possible difference in the overall immune investment between the castes either in the absence or presence of infection, we performed a combined z-transformation of all worker and queen pupae together, calculated the mean normalised gene expression per pupa as above and carried out Wilcoxon Rank-Sum tests to test for a different immune investment of the castes in (i) the absence (combined I-W- and I-W + ) and (ii) the presence (combined I + W- and I + W + ) of infection, and corrected for multiple testing.

**Infection progression.** To assess how pathogen load progressed over the course of infection for the pupae of both castes, we ran separate linear models (LMs) for worker and queen pupae, testing for the effect of infection period on the infection load. We controlled for worker presence in both worker and queen pupae models (Supplementary Tables S9a, b) and as above, for population in the model for the worker pupae only (Supplementary Table S9a). Following a significant interaction between infection period and load for both queen and worker pupae, we performed all-pairwise post hocs on whether the pupal infection loads differed between the early, middle and late stages of infection period using

the 'emmeans' package (vs 1.10.0)[77] (Supplementary Tables S9a, b). Prior to analysis, we obtained normality of the data distribution by transforming the data using the ordered quantile normalisation.

**Worker reaction to extract application.** To test whether the application of signal extract triggered higher unpacking of the otherwise healthy treated pupae than application of the non-signal extract, we ran separate binomial Generalised Linear Models (GLMs) with a binomial error distribution and a logit link function for worker and queen pupae. The response variable, unpacking, was modelled as a binary outcome, while extract type (signal vs non-signal) was included as the sole explanatory variable while controlling for experimental block, extract batch and in the case of worker pupae also population (Supplementary Table S10). We checked for overdispersion by comparing the ratio of residual deviance to degrees of freedom. We performed one-sided hypothesis tests, as we were specifically testing for an increase in unpacking when the extract came from signalling worker pupae.

### Reporting summary

Further information on research design is available in the Nature Portfolio Reporting Summary linked to this article.

## Data availability

Raw data of the integrated $^{12}$C and $^{13}$C areas of all samples, as well as the mass spectra of the native CHCs (Supplementary Table S13) have been deposited in the ISTA Research Explorer (ISTA REx) repository under https://doi.org/10.15479/AT-ISTA-20471. Material requests should be addressed to Sylvia Cremer. Source data are provided with this paper.

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

## Acknowledgements

We thank Joergen Eilenberg and Nicolai V. Meyling for the fungal strain, and the ISTA Social Immunity team, Jonghyun Park and Yuko Ulrich for ant collection. We also thank the Social Immunity team, in particular David Moreno Martínez, Tanvi Madaan, Wilfrid Jean Louis and Jessica Kirchner, for experimental and molecular support, as well as Friedrich Fochler for technical support with the chemical analysis, and the ISTA Lab Support Facility, including the mass spectrometry unit, for general and chemical laboratory support. We further thank Marco Ribezzi for advice on $^{13}$C calculations and Ernst Pittenauer for discussion of the chemical data, Chris Pull and Michael Sixt for project discussion, and the Social Immunity team for comments on the manuscript. The study was funded by the European Research Council (ERC) under the European Union's Horizon 2020 research and innovation Programme (No. 771402; EPIDEMICSonCHIP) to SC.

## Author contributions

E.H.D. and S.C. conceived the study with input by T.S.; E.H.D., M.H., N.K., A.V.G., L.L., J.R., F.B., F.S., H.L., H.R. and S.C. performed the experiments generating the raw data, which were analysed by M.H., N.K., A.V.G., L.L. and T.S.; E.H.D. and A.V.G. curated the data; E.H.D. performed the statistical data analysis; figures were created by E.H.D. and A.V.G. after joint conceptualisation with S.C.; E.H.D. and S.C. wrote the paper with support of M.H., N.K., A.V.G., L.L. and T.S.; all authors approved the final manuscript version. Funding was obtained by S.C.

## Competing interests

The authors declare no competing interests.
