## [Transparent Peer Review file · Nature Communications]

Altruistic disease signalling in ant colonies

Corresponding Author: Professor Sylvia Cremer

Version 0:

Reviewer comments:

Reviewer #1

(Remarks to the Author)

The elegant study by Dawson is based on the interesting result that cuticular hydrocarbons (CHCs) of developing ants provide cues on their infection with a fungus that are perceived by adult ants that then destroy the pupae (Pull et al. 2018). Here, the authors report that these cues are active signals, leading to altruistic suicide, based their new results that the worker pupae only signal when adult workers are present and that the signals do not correlate with infection load. Furthermore, the authors show that the reproductive female caste, queens, do not change their CHCs and are simultaneously able to contain the fungus, even at higher infection doses. They also demonstrate that attending workers are not the source of these CHCs. Thus, they can conclude that the CHC changes are produced by the worker pupae. Unfortunately, I am not convinced that the results can rule out that the CHC changes are a passive consequence of the immune system of the ants. In fact, the authors make an argument that this could be the case by citing other studies. Immune system differences could also explain the lack of changes in queen pupae. Therefore, additional data on the immune system activity is needed in workers and queens under the experimental conditions to substantiate the claim that “infected ant pupae [...] actively emit a chemical signal” (line 16). Hopefully, those samples are still available for such analyses.

Even though this work is based on previously published identification of the CHC compounds tritriacontene and tritriadontadiene, these compounds are in need of a more accurate characterization to make comparisons to other studies (e.g., Wagoner et al. 2020) possible and allow exact replication of this study. Specifically, these compounds can occur in several isoforms (Pull et al. 2018 seem to have found two for tritriacontene) and I do not see which ones were studied here.

Another methodological concern is the separate assessment of queen and worker pupae in 2019 and 2021 respectively. In addition, the queens stem only from one population, while workers from two populations were included.

Otherwise, the manuscript is excellent and the study conducted quite well. The sample sizes are very good, the methods sophisticated and described thoroughly, the supplemental data are helpful, and the findings could certainly be very interesting and relevant.

Minor comments:

Line 33: Can you support this statement with a citation or two?

Line 46: ...increasing “its” own... ?

Line 56: “if signaling promotes group health” should be listed as an independent factor because its sets apart cases of communicable vs. non-communicable diseases, which is significant.

Line 73: Please write out the scientific names of these compounds.

Caption of Figure 1A: To make it clearer, I would suggest “...as well as the pathogen free control pupae with (I-W+; dark pink) and without (I-W-) workers (Tables S1,S2).

Reviewer #2

(Remarks to the Author)

This is a nice study that tackles a fascinating topic: how species with complex social groups have evolved mechanisms for controlling infectious disease.

The experiments here show that worker pupae emit two specific hydrocarbons when infected with a model pathogen, but

only in the presence of adult workers who can respond to the signals. The response of workers to these signals is “unpacking” behavior, in which the (infected) developing pupae are removed from their pupal cases and destroyed. When adult workers are not present, the pupal signals are not evident. Queen pupae, who can more effectively fight off the infection, do not signal in this way. An especially nice component of this research is the use of isotopically-labeled worker hydrocarbons, which allowed the researchers to identify the source of the hydrocarbons, and exclude the possibility that workers were depositing the hydrocarbons on the pupae. The results here appear to provide confirmation of patterns seen in previous studies of both ants and honey bees.

Although this work advances beyond what was previously reported in Pull et al (2018), there is also some overlap. Using the same experimental organisms and some of the same procedures, Pull et al reported (among other things) the unpacking of infected worker pupae by adult workers and the identity of four hydrocarbons that were correlated with this behavior, including the two implicated in this study. In this study, the authors have performed similar experiments, but have also added in the useful comparison to infected queen pupae, the isotopic clarification of the hydrocarbon origins.

My biggest concern is the lack of a manipulative experiment showing the function of the dienes, C33:1 and C33:1. It should be straightforward to select uninfected worker pupae, treat some with these hydrocarbons (individually and in combination) and test whether this induces significantly higher unpacking behavior by the attendant workers (compared to control pupae who are treated with putative non-signaling hydrocarbons). Going even further, it would also be interesting to see if unpacking of queen pupae can be induced by addition of these hydrocarbons.

More minor questions and comments:

The procedure of exposure to *Metarhizium*, by rolling the pupae in a fungal suspension of the given concentration, would seem to add an extremely high dosage of spores. In addition, the post-exposure isolation for three days without the benefit of worker grooming would further intensify the exposure. How does this procedure reflect real-world exposures to pathogens?

The authors describe exclusion of some replicates due to poor pupal or larval condition (e.g. damage, death, etc). These exclusions seem justified, but for transparency the authors should indicate the initial numbers and the numbers excluded for each reason in the supplementary material or online repository.

Raw data and/or representative profiles from the GC-MS should be provided or archived in an available database.

For the experiment in which sampling times were divided into early, middle, and late: why was the late period 12 hours long, versus six for the early and middle? Shouldn't the late period be defined as the 30-36 hour period, not 30-42 hours?

Reviewer #3

(Remarks to the Author)

This work represents a nice and novel contribution to the field of socio-eco-immunology. The research combines molecular techniques, chemical analyses, immunological measurements and behavioral observations to decipher the role that ant pupae have on colony-wide disease resistance. Based on their data, we can now better visualize the nuanced dynamics between the different castes and different developmental stages within an ant colony. No longer can we assume that pupae in the social Hymenoptera represent an inactive and unresponsive developmental stage when its natal colony faces disease risks. I find this original work well conceived/planned, well analyzed and written. In particular, I find their methodology for disentangling the origin of cuticular hydrocarbons (either from the tending workers or from the worker-pupae or queen-pupae) ingenious. Using the $^{12}\text{C}:^{13}\text{C}$ ratio allowed the researchers to identify the origin of two immune-associated cuticular hydrocarbons C33:1 and C33:2 when ants are facing fungal infection. They established that C33:1 and C33:2 infected worker-pupae (which are less immunocompetent and have little future reproductive potential—as they are sterile when mature) produce distress hydrocarbons that elicit their destruction by the tending workers. In contrast, the fungal infected queen-pupae (which appear to have stronger immunocompetence than the worker-pupae and have high future reproductive potential) do not increase the production of these signals and therefore, are less likely to be targeted for destruction by the tending workers. This work merits to be published. Yet, I have a few suggestions that should probably be considered.

1) Overall, this work is well written. Yet, not all readers necessarily know much about ant biology. So additional basic information would be useful for those readers that are not experts in the field of social insect biology. Explaining that this ant species has a cocoon while other ant species do not (naked pupae) would be important. Furthermore, explain that the pupae developmental stage is solely dependent on workers, that it does not feed, that it is immobile, that it transforms itself inside the cocoon from etc. This additional basic information would help the readers realize that in spite such characteristics, the pupae can still perceive the presence of fungal conidia, be immunocompetent and participate not only in chemical communication with adult nestmates but also exhibit altruistic behavior at this immature stage of development! And that is cool!

2) Page 2 lines 42-43: there is no citation for the vibratory displays in termites (Pathogen alarm responses by Rosengaus) or their heightened frequency of allogrooming behavior when exposed to *Metarhizium* (several articles by Rosengaus as well.

- Rosengaus, R.B., Lefebvre, M.L., Jordan, C. and Traniello, J.F.A. (1999) Pathogen alarm behavior in a termite: A new form of communication in social insects. *Naturwissenschaften*, 86:544-548.
- Rosengaus, R.B., Maxmen A.M., Coates, L.E. and Traniello, J.F.A. (1998) Disease resistance: a benefit of sociality in the dampwood termite *Zootermopsis angusticollis* (Isoptera: Termitidae). *Behavioral Ecology and Sociobiology*, 44:125-134.
- Rosengaus, R.B. and Traniello, J.F.A. (2001) Disease susceptibility and the adaptive nature of colony demography in the dampwood termite *Zootermopsis angusticollis*. *Behavioral Ecology and Sociobiology*, 50(6):546-556.

- 3) Page 4, line 87: be sure to explain that such chemicals are not being ingested by the pupae, but are just spread/shared through social interactions whenever the workers are grooming, moving the pupae around the nest etc.). Again, not everyone knows much about ant pupae and they may think you are referring to ingestion of the CHC?
- 4) Page 7, line 146: "feeble"??Perhaps a better descriptor would be "less immuno-competent"?
- 5) Page 7, line 150: perhaps add "destructive signaling"?
- 6) Page 8, line 174: as described, I immediately visualized this as a "naked pupae" species, only to later read that pupae in this species have a cocoon. Can you really see the eyes through the cocoon?
- 7) What are the different symptomologies as time after exposure progresses in the pupae? Do you see any differences? What about the symptoms of disease for the tending ants? do they respond to the conidia on the pupae? I know the workers lick/groom the pupae more.. But do the grooming ants get infected themselves? What is their symptomatology?
- 8) Page 10; line 231: In what ways do they appear damaged or dead? How do you tell that pupae are dead if they are surrounded by a cocoon?
- 9) Throughout the ms: the authors use: "in dependence". That is awkward. Perhaps replace that with "depending on" or "as a function of"
- 10) I believe the researchers were diligent in their conservative inclusion of data into their analyses.
- 11) Page 18, line 417: perhaps "chemical bouquet"?

A pleasure reading your work.. [editorial note: reviewer name redacted]

Version 1:

Reviewer comments:

Reviewer #1

(Remarks to the Author)

I would like to congratulate the authors for the very thorough revision, which included additional data that addressed all my concerns.

Reviewer #2

(Remarks to the Author)

I was happy to see this revised version of the previous manuscript. I appreciate that the authors addressed the major criticisms from the previous review, both mine and those of the other reviewers. The clarifications and added experiments significantly improve and strengthen this study, and I have no additional major comments. This is really a fascinating system and I look forward to seeing continue research along these lines in this and other model systems.

Minor comments:

Line 50 – "aggressed" is a bit awkward. Better to say "attacked" or "subjected to aggression by".

Line 122 – Better to say "before the experiment" rather than "ahead of the experiment".

Line 174 – Change "where, as previously shown" to "when, as previously shown". (and remove comma before "where").

Use of commas is inconsistent throughout and would benefit from input from style editor.

POINT-BY-POINT RESPONSE

Dear Editor, dear Reviewers,

We would like to thank you for your thorough reviews and the detailed suggestions on how to improve our study.

We have addressed all the points raised, as detailed below, and are pleased to submit a revised version of the manuscript that includes novel experiments and analyses.

We believe that having examined the suggested points in greater detail has provided not only deeper insights into the interplay between individual immunity and chemical signalling in infected ant brood, and the chemical complexity of the signal itself, but also a clearer understanding of how disease detection and communication may have evolved across social insects.

In the revised version, we provide all created data as source data with the manuscript and supplement.

Thank you again for your time and efforts.

With kind regards,
Erika H. Dawson and Sylvia Cremer
corresponding authors, on behalf of all authors

REVIEWER COMMENTS

Reviewer #1 (Remarks to the Author):

The elegant study by Dawson is based on the interesting result that cuticular hydrocarbons (CHCs) of developing ants provide cues on their infection with a fungus that are perceived by adult ants that then destroy the pupae (Pull et al. 2018). Here, the authors report that these cues are active signals, leading to altruistic suicide, based their new results that the worker pupae only signal when adult workers are present and that the signals do not correlate with infection load. Furthermore, the authors show that the reproductive female caste, queens, do not change their CHCs and are simultaneously able to contain the fungus, even at higher infection doses. They also demonstrate that attending workers are not the source of these CHCs. Thus, they can conclude that the CHC changes are produced by the worker pupae.

R1.1

Unfortunately, I am not convinced that the results can rule out that the CHC changes are a passive consequence of the immune system of the ants. In fact, the authors make an argument that this could be the case by citing other studies. Immune system differences could also explain the lack of changes in queen pupae. Therefore, additional data on the

immune system activity is needed in workers and queens under the experimental conditions to substantiate the claim that “infected ant pupae [...] actively emit a chemical signal” (line 16). Hopefully, those samples are still available for such analyses.

We thank the reviewer for raising this point, as we agree that our previously provided data could not rule out that the presence of workers may have some influence on individual immunity, which could directly link to the changes in CHC profile. All the pupae in the CHC experiment were extracted for their DNA to obtain fungal infection load (qPCR of fungal DNA), hence immune gene expression analysis requiring RNA could not be obtained from the same samples.

We would like to clarify that we had previously repeated the experiment with worker pupae to analyse the expression of three immune genes acting across different stages of the immune response. However, we did not include these data in the initial manuscript version, since they revealed no differences between pupae with and without workers in either the infected or control groups, which we considered a trivial finding at the time. It was only through the reviewer’s comment that we recognised the importance of these data in supporting our conclusions that the chemical profile of the ants is actively modulated by the ants themselves, rather than being a passive consequence of immune changes that might directly change their scent. This conclusion can be supported by the data, as the CHC profiles do not align with the gene expression patterns. Specifically, the chemical signal peaks were only upregulated in infected pupae with workers being present, whereas immune gene expression – consistently across all genes (see Fig. 1c & S3a, Table S3) – depended solely on infection status, showing no effect of worker presence/absence (lines 139-150).

We have now repeated the experiment also for queen pupae and found the exact same immune gene expression pattern as in the worker pupae: an upregulation of immune gene expression upon infection, independent of worker presence or absence (Fig. 1d, S3b and Table S8). Again, these immunity data cannot explain the CHC patterns, which show no differences across the groups (lines 166-171).

Therefore, analysis of physiological immunity of both castes revealed that the chemical signalling is not a mere representation of the immune state of the pupae, allowing us to conclude that the chemical signalling follows a mechanism independent of switching on the immune response.

The reviewer’s suggestion to compare between castes revealed an additional important aspect: our data indicate that baseline immunity is higher in queen pupae (even independent of their larger body size, since immune gene expression values are normalised to a housekeeping gene), whereas worker pupae are able to reach similar expression levels upon infection. Previous work has shown that a high baseline immune investment is crucial for mounting a fast defence against pathogens, effectively reducing infection progression. This may explain why queen pupae are better able to control infection than worker pupae. Lastly, while it was known that social insect queens exhibit stronger physiological immunity than workers, this information was limited to adult stages. As also noted by Rev3, the fact that pupae can mount such responses during metamorphosis highlights another exciting finding of our study (lines 171-179).

R1.2

Even though this work is based on previously published identification of the CHC compounds tritriacontene and tritriadontadiene, these compounds are in need of a more accurate characterization to make comparisons to other studies (e.g., Wagoner et al. 2020) possible and allow exact replication of this study. Specifically, these compounds can occur in several isoforms (Pull et al. 2018 seem to have found two for tritriacontene) and I do not see which ones were studied here.

We appreciate this comment by the reviewer and have performed DMDS-derivatisation followed by isomer characterisation of the C33:2 and C33:1. This indeed gave very interesting additional insight, both in how the ants regulate their signalling and to which degree the chemical profiles of sick ants and bees can be compared.

The reviewer is correct in pointing out that our native GC-MS chromatograms showed two separately eluting C33:1 peaks, one being upregulated in unpacked pupae, the other not (as already established in Pull et al *eLife* 2018). Previous work on *Lasius neglectus* adult workers (Cremer et al *PLoS One* 2008) had also described the presence of five C33:1 isomers and suggested the presence of several, still unidentified, C33:2 isomers. Our isomer analysis identified a total of eight C33:2 isomers and six C33:1 isomers being present in the pupal samples (Figs. S4,S5).

We were excited to discover that one of the C33:1 isomers present in our ants – 10-C33:1 – which had previously been identified in honeybees where it is upregulated in sick brood, synthesized in its natural (Z)-configuration, and was found to trigger hygienic behaviour by the bees when applied in its pure form to healthy brood (Wagoner et al 2020). This isomer overlap between ants and bees allowed us to conclude that our isomers are also present in the (Z)-configuration. Even if this was already likely since the (Z)-configuration is the only reported stereoisomeric configuration of naturally-occurring insect CHCs, direct retention time overlap of the synthetic (Z)-10-C33:1 with our pupal extract (Fig. S6) confirmed this (lines 258-265).

We also used the synthetic (Z)-10-C33:1 in our bioassay (see also response to Rev2), to test if it had a similar function in the ants as in honeybees, where it is an important indicator for viral infection and an elicitor of hygienic behaviour (Wagoner et al 2002). Yet, three lines of evidence suggest that this isomer does not play an important role in fungal disease signalling in ants: (i) application of the synthetic compound did not elicit pupal unpacking in our experiments (while natural extract transfers did, see below and Fig. 2), and (ii) (Z)-10-C33:1 was the isomer of the lowest abundance of C33:1 in the native GC-MS chromatograms, which (iii) did not differ in its relative proportion between signalling and non-signalling pupae (Table S12; lines 265-272).

Interestingly, however, the two peaks (C33:2 and C33:1) differ in the way the signal is encoded: C33:2 is increased approx. 20 times in signalling pupae, with the eight isomers contributing in similar proportions throughout. C33:1, on the other hand, consists of six isomers, five of which co-elute in a cluster (13- to 9-C33:1), and the other as a separate peak

(7-C33:1). These are in line with the first peak (RI 3279) being identified by Pull et al 2018 and confirmed in this study to be increased in overall abundance in the signalling pupae, and the second peak (RI 3288; Table S13), which is not upregulated in unpacked pupae). Interestingly, the most abundant compound of the unpacking-relevant cluster (12-C33:1) reduces drastically in signalling pupae, while two others (13- and 11-C33:1) double in their proportion. The remaining two (10- and 9-C33:1) don't show any substantial changes. Since the overall abundance of the C33:1 isomer cluster is only 3-times higher in the signalling pupae, it seems like the main information here lies in the proportional change of the constituting isomers (lines 224-243).

Therefore, as suggested by the reviewer, only the detailed analysis at the level of isomers allowed us to understand the complexity of the chemical signal and clearly revealed that ants and bees have not simply coopted the same compounds to be relevant disease cues or signals. Instead, the observed parallel evolution across social insect taxa reveals the importance of disease detection and colony-level defence lines in their colonies (lines 274-288).

R1.3

Another methodological concern is the separate assessment of queen and worker pupae in 2019 and 2021 respectively. In addition, the queens stem only from one population, while workers from two populations were included.

We are aware that it would have been optimal to also have been able to study the queen pupae from different populations, yet this was not possible due to sampling constraints. Whereas worker pupae are produced over a long period under summer conditions in the field and laboratory, queen pupae are only produced once a year in a single cohort, whilst they are only produced very rarely under laboratory conditions. For our experiments, we therefore relied on being able to collect queen pupae of the exact right stage in the field (not too far developed as otherwise they would be helped by the workers to exit their cocoons as part of normal hatching, whereas unpacking in the context of destructive disinfection occurs prematurely when pupae are not yet ready to hatch). This was only possible for the Jena population, where we also received support by a local ant group (Yuko Ulrich's group) to detect the presence of the pupae and to collect them at the right stage.

However, since Pull et al had already established that destructive disinfection is robust across populations of *L. neglectus* and even across species (*L. niger*), and because our new experiments further confirm this robustness across multiple years (Pull et al and this study, which now also includes multiple replicates over several years), we are confident in the generalisability of our findings across populations and time.

R1.4

Otherwise, the manuscript is excellent and the study conducted quite well. The sample sizes are very good, the methods sophisticated and described thoroughly, the supplemental data are helpful, and the findings could certainly be very interesting and relevant.

Thank you very much for this positive feedback on our work.

R1.5

Minor comments:

Line 33: Can you support this statement with a citation or two?

We added Locatello et al 2013 and Gormally et al 2022 for the concealing on the infection state in other species (line 53).

Line 46: ...increasing “its” own... ?

Added (line 66).

Line 56: “if signaling promotes group health” should be listed as an independent factor because its sets apart cases of communicable vs. non-communicable diseases, which is significant.

We now make this specific (lines 74/75).

Line 73: Please write out the scientific names of these compounds.

We added the UPAC names (lines 101/102).

Caption of Figure 1A: To make it clearer, I would suggest “...as well as the pathogen free control pupae with (I-W+; dark pink) and without (I-W-) workers (Tables S1,S2).

We clarified (line 1224).

Reviewer #2 (Remarks to the Author):

This is a nice study that tackles a fascinating topic: how species with complex social groups have evolved mechanisms for controlling infectious disease.

The experiments here show that worker pupae emit two specific hydrocarbons when infected with a model pathogen, but only in the presence of adult workers who can respond to the signals. The response of workers to these signals is “unpacking” behavior, in which the (infected) developing pupae are removed from their pupal cases and destroyed. When adult workers are not present, the pupal signals are not evident. Queen pupae, who can more effectively fight off the infection, do not signal in this way. An especially nice component of this research is the use of isotopically-labeled worker hydrocarbons, which allowed the researchers to identify the source of the hydrocarbons, and exclude the possibility that workers were depositing the hydrocarbons on the pupae. The results here appear to provide confirmation of patterns seen in previous studies of both ants and honey bees.

Although this work advances beyond what was previously reported in Pull et al (2018), there is also some overlap. Using the same experimental organisms and some of the same procedures, Pull et al reported (among other things) the unpacking of infected worker pupae by adult workers and the identity of four hydrocarbons that were correlated with this behavior, including the two implicated in this study. In this study, the authors have

performed similar experiments, but have also added in the useful comparison to infected queen pupae, the isotopic clarification of the hydrocarbon origins.

R2.1

My biggest concern is the lack of a manipulative experiment showing the function of the dienes, C33:1 and C33:1. It should be straightforward to select uninfected worker pupae, treat some with these hydrocarbons (individually and in combination) and test whether this induces significantly higher unpacking behavior by the attendant workers (compared to control pupae who are treated with putative non-signaling hydrocarbons). Going even further, it would also be interesting to see if unpacking of queen pupae can be induced by addition of these hydrocarbons.

We agree with the reviewer that the evidence provided in the first submission was only correlational, and that manipulative experiments are required to establish a causal relationship. We therefore developed a bioassay in which we experimentally transferred the CHC extracts of signalling worker pupae, as well as that of non-signalling worker pupae as a control onto healthy worker pupae and queen pupae. This allowed us to test whether transfer of the chemical signal alone was sufficient to induce unpacking by the workers. Indeed, application of signal extract induced unpacking, which was not only true for receiving healthy worker pupae (Fig. 2a, lines 201-207), but even when the extract from the signalling worker pupae was transferred between castes, to the queen pupae (Fig. 2b, lines 207-210). We further show that our procedure by itself was not eliciting any unpacking (application of solvent only, line 746 ff), and that the transfer of non-signal extract only elicited little worker unpacking (Figs. 2a,b). We can therefore conclude that it is the *change* in the chemical signatures from non-signalling to signalling pupae (visualised in Figs. 2c,d) that by itself leads to trigger unpacking.

We are aware that this change constitutes a change in CHC blend and also recognise the reviewer's further suggestion to test the effects of the signal CHCs not only in combination as they occur in the natural extracts, but also separately. However, our detailed analysis of the C33:2 and C33:1 peaks revealed the existence of at least 14 different isomers of the two CHCs, which show a complex pattern of up- or downregulation in their relative proportions (see detailed in our response to R1.2). Reconstructing and applying synthetic blends that accurately reproduce the diversity of isomers and the change of their relative abundances of such complexity as observed in our natural signal profiles would be technically extremely challenging and beyond the scope of the current study, especially since the synthesis of alkadienes can involve highly demanding analytical chemistry. Moreover, the observed complexity in the C33:2 and C33:1 patterns is consistent with findings in other recognition systems of social insects, which often rely on diverse compound ratios rather than single-molecule abundances. Therefore, targeted manipulation of individual isomers on their own may not be straightforward to enable us to get a more detailed insight into the biological nature of the signal given the natural profiles change in 20-fold upregulation of our eight C33:2 isomers (in equal proportion) and the 3-fold overall abundance increase with changing isomer ratios in at least 3 abundant isomers.

Overall, our experiments revealed that the change in chemical blend alone is sufficient to trigger unpacking, and that the workers would react to similar changes also in the queen pupae, in case they would produce them.

R2.2

More minor questions and comments:

The procedure of exposure to *Metarhizium*, by rolling the pupae in a fungal suspension of the given concentration, would seem to add an extremely high dosage of spores. In addition, the post-exposure isolation for three days without the benefit of worker grooming would further intensify the exposure. How does this procedure reflect real-world exposures to pathogens?

In Pull et al. 2018, we quantified by use of quantitative real time PCR targeting the fungal DNA, that the application dose of 1 μ l of a 1×10^6 spores/ml fungal suspension results in approx. 1800 spores attaching to the pupal cocoon of a worker pupae, of which 95 spores reach the pupa inside the cocoon, where contact with the cuticle induces germination as the first step of infection. Here, we used the same dose for the worker pupae, but exposed the queens to a higher load, which we inferred to lead to an approximate load of 940 spores reaching a queen pupa (lines 355-358). This higher dose was chosen because queen pupae are of much larger size than worker pupae and preliminary work showed that an approx. half of this dose was insufficient to successfully induce an infection in the queen pupae (lines 364/365). Like other entomopathogenic fungi, *Metarhizium* spores are highly abundant in nature: single cadavers show outgrowth of spore packages making up to 12 million spores and soil can contain 5000 spores / gram soil (lines 358-360). Therefore, we consider our exposure doses to be within natural range.

To standardise our experimental procedures and ensure a high probability of infection, we introduced an isolation step to prevent workers from grooming off the spores after application. Yet, we would like to emphasise that destructive disinfection itself does not rely on this experimental setup as shown in Fig. 1a of Pull et al 2018. Importantly, *L. neglectus* naturally separates its brood according to developmental stage, creating brood chambers that consist exclusively of pupae, separate from larval chambers. Reflecting the reduced care requirement of the pupae (i.e. pupae are not fed, unlike larvae), we also generally observe fewer workers present in these pupal chambers in our stock colonies. While our three-day isolation may not precisely replicate nest conditions, we do not consider it to introduce an unrealistic scenario.

More broadly, the fact that destructive disinfection has evolved as a targeted response to fungal infection suggests that established infections in pupae must occur with sufficient frequency under natural conditions to impose selective pressure. Given the ants build their nests in the soil, which is rich in infectious stages of entomopathogens (see above), exposure of the brood is likely to occur frequently. While such early-stage pathogen exposures are likely often removed through allogrooming before they can progress, it follows that grooming is not always successful or fast enough, allowing infections to occasionally establish and trigger the destructive disinfection response. Thus, our use of isolation served to standardise

infection success under experimental conditions, but does not undermine the ecological or evolutionary relevance of destructive disinfection.

R2.3

The authors describe exclusion of some replicates due to poor pupal or larval condition (e.g. damage, death, etc). These exclusions seem justified, but for transparency the authors should indicate the initial numbers and the numbers excluded for each reason in the supplementary material or online repository.

We now provide all these numbers in the respective methods sections (CHC and infection progression experiment, lines 390-400; immune gene experiment, lines 589-92; bioassay, lines 713-14) and in the data exclusion section of the reporting summary.

R2.4

Raw data and/or representative profiles from the GC-MS should be provided or archived in an available database.

We have now included representative chromatograms of the signalling vs non-signalling extracts in the new main Fig. 2c,d, as well as in Suppl. Fig. S6. We also provide the isomer structures in Suppl. Fig. S4, their mass spectra in Suppl. Fig. S5, the and the isomer ratios in Suppl. Tables S11 and S12. Our source data further provide the pupal relative peak abundances for all 18 analysed peaks for each pupa, as well as the EIC abundances of all C33:2 and C33:1 isomers.

R2.5

For the experiment in which sampling times were divided into early, middle, and late: why was the late period 12 hours long, versus six for the early and middle? Shouldn't the late period be defined as the 30-36 hour period, not 30-42 hours?

Most unpacking occurs at time points 12 and 18 hours, so that these time points received higher sampling effort than the later time points, when unpacking becomes rarer. Therefore, to balance our sample sizes and ensure higher statistical power, we combined the last three sampling time points to a single late period (lines 891-94).

Reviewer #3 (Remarks to the Author):

This work represents a nice and novel contribution to the field of socio-eco-immunology. The research combines molecular techniques, chemical analyses, immunological measurements and behavioral observations to decipher the role that ant pupae have on colony-wide disease resistance. Based on their data, we can now better visualize the nuanced dynamics between the different castes and different developmental stages within an ant colony. No longer can we assume that pupae in the social Hymenoptera represent an

inactive and unresponsive developmental stage when its natal colony faces disease risks. I find this original work well conceived/planned, well analyzed and written. In particular, I find their methodology for disentangling the origin of cuticular hydrocarbons (either from the tending workers or from the worker-pupae or queen-pupae) ingenious. Using the 12C:13C ratio allowed the researchers to identify the origin of two immune-associated cuticular hydrocarbons C33:1 and C33:2 when ants are facing fungal infection. They established that C33:1 and C33:2 infected worker-pupae (which are less immunocompetent and have little future reproductive potential—as they are sterile when mature) produce distress hydrocarbons that elicit their destruction by the tending workers. In contrast, the fungal infected queen-pupae (which appear to have stronger immunocompetence than the worker-pupae and have high future reproductive potential) do not increase the production of these signals and therefore, are less likely to be targeted for destruction by the tending workers. This work merits to be published. Yet, I have a few suggestions that should probably be considered.

R3.1

1) Overall, this work is well written. Yet, not all readers necessarily know much about ant biology. So additional basic information would be useful for those readers that are not experts in the field of social insect biology. Explaining that this ant species has a cocoon while other ant species do not (naked pupae) would be important. Furthermore, explain that the pupae developmental stage is solely dependent on workers, that it does not feed, that it is immobile, that it transforms itself inside the cocoon from etc. This additional basic information would help the readers realize that in spite such characteristics, the pupae can still perceive the presence of fungal conidia, be immunocompetent and participate not only in chemical communication with adult nestmates but also exhibit altruistic behavior at this immature stage of development! And that is cool

Thank you for pointing this out. We have added this basic information in the introduction (lines 93 ff), as well as in the Results section, where we now also discuss our new analyses of the pupal immune genes (lines 174-77). We agree that it may not be expected from an organism undergoing metamorphosis to have such fine-tuned responses.

R3.2

2) Page 2 lines 42-43: there is no citation for the vibratory displays in termites (Pathogen alarm responses by Rosengaus) or their heightened frequency of allogrooming behavior when exposed to *Metarhizium* (several articles by Rosengaus as well).

- Rosengaus, R.B., Lefebvre, M.L., Jordan, C. and Traniello, J.F.A. (1999) Pathogen alarm behavior in a termite: A new form of communication in social insects. *Naturwissenschaften*, 86:544-548.
- Rosengaus, R.B., Maxmen A.M., Coates, L.E. and Traniello, J.F.A. (1998) Disease resistance: a benefit of sociality in the dampwood termite *Zootermopsis angusticollis* (Isoptera: Termopsidae). *Behavioral Ecology and Sociobiology*, 44:125-134.
- Rosengaus, R.B. and Traniello, J.F.A. (2001) Disease susceptibility and the adaptive nature

of colony demography in the dampwood termite *Zootermopsis angusticollis*. Behavioral Ecology and Sociobiology, 50(6):546-556.

We are sorry for our confusing referencing. We had previously only cited the connection of the vibratory signal to grooming providing a reference at the very end of the sentence, intermingled with the wound care in ants. We now provide a reference describing the vibrational signal and one revealing its grooming-inducing effect on nestmate termites, and cite the references throughout the text for clarity (lines 62-64).

R3.3

3) Page 4, line 87: be sure to explain that such chemicals are not being ingested by the pupae, but are just spread/shared through social interactions whenever the workers are grooming, moving the pupae around the nest etc.). Again, not everyone knows much about ant pupae and they may think you are referring to ingestion of the CHC?

In addition to now spelling out that the pupae do not engage in feeding (lines 117, 174-77), we have now specified that we needed to exclude external transfer by grooming (lines 115-120). We also extended this sentence to mention that the pupal and adult CHC profiles overlap and referenced the according publications.

R3.4

4) Page 7, line 146: “feeble”??Perhaps a better descriptor would be “less immunocompetent”?

We have exchanged the “feebler worker immune system” by “less potent” and contrast it to a “stronger” queen immune system (lines 189-90).

R3.5

5) Page 7, line 150: perhaps add “destructive signaling”?

We have expanded to “signalling for own destruction” (line 194).

R3.6

6) Page 8, line 174: as described, I immediately visualized this as a “naked pupae” species, only to later read that pupae in this species have a cocoon. Can you really see the eyes though the cocoon?

Yes, the eyes and mandibles are visible through the worker cocoon, with some magnification, as shown in the below photograph that was taken by my previous PhD student CD Pull, contrasting a pupa with and one with removed cocoon. We determine the stage of each worker pupae before the experiment by use of a stereoscope, as it is crucial for our experiments that the pupae have not yet completed their metamorphosis to become young hatching workers (callows) during the whole period of the experiment (3 days of incubation and up to 42 h in the experiment). Natural hatching would interfere with our destructive

disinfection unpacking observations, as the workers also support the pupae from leaving the cocoon then, but, if it occurs, it can easily be disentangled as those then emerge as walking adults.

For the queen pupae, however, the cocoon is not transparent and we need to rely on a comparative approach here as all queen pupae produced in a colony at a time belong to the same age cohort, so that opening some queen cocoons gives us a very good estimate for the remaining pupae. We now explain in the manuscript (lines 324-3277): “Experiments were performed with pupae of a standardised developmental stage (white pupae with black eyes, as visible through the cocoon using a stereoscope (Leica S6 E) for worker pupae; for queen pupae, the stage can be reliably inferred by checking some pupae in the colony, as they develop highly synchronised in same-age cohorts).”

Photo C D Pull. Arrow towards the eye that is visible through the cocoon.

R3.7

7) What are the different symptomologies as time after exposure progresses in the pupae? Do you see any differences? What about the symptoms of disease for the tending ants? do they respond to the conidia on the pupae? I know the workers lick/groom the pupae more.. But do the grooming ants get infected themselves? What is their symptomatology?

Thank you for pointing this out. After exposure to the fungal conidiospores (referred to “spores” throughout), it takes some time for the spores to germinate, form an appressorium, and penetrate the host cuticle. Once internal infection is established, spores are no longer externally transmissible. During the following days of internal infection, the pathogen replicates inside the host and kills it. Infectiousness resumes only after host death, when hyphae emerge and sporulation begins. In Pull et al 2018, we showed that destructive disinfection occurs before this infectious stage, posing no risk to workers. We have clarified this in the manuscript (lines 97-99): “Notably, this destructive disinfection is performed by the workers during the non-infectious incubation period of the fungal pathogen, therefore not inducing any disease-risk to the workers.”

R3.8

8) Page 10; line 231: In what ways do they appear damaged or dead? How do you tell that pupae are dead if they are surrounded by a cocoon?

As seen in the above photograph, the cocoon allows partial visual inspection of the pupa inside, and dead pupae often show a change in colouration. In addition, the cocoon itself becomes less oval and takes on a 'shrivelled' appearance. We now describe this in the manuscript as: "as identified by a 'shrivelled' morphology or a change in colouration" (lines 390-92). To ensure that we analysed only actively signalling individuals, we conservatively excluded any brood showing signs of impairment or death.

R3.9

9) Throughout the ms: the authors use: "in dependence". That is awkward. Perhaps replace that with "depending on" or "as a function of"

We have changed this throughout.

R3.10

10) I believe the researchers were diligent in their conservative inclusion of data into their analyses.

Thank you.

R3.11

11) Page 18, line 417: perhaps "chemical bouquet"?

Changed (line 574).

A pleasure reading your work.. [**editorial note: reviewer name redacted**]

We would like to thank all reviewers again for their careful assessment of our work and the detailed suggestions for improvement.

POINT-BY-POINT RESPONSE

REVIEWERS' COMMENTS

Reviewer #1 (Remarks to the Author):

I would like to congratulate the authors for the very thorough revision, which included additional data that addressed all my concerns.

Thank you very much.

Reviewer #2 (Remarks to the Author):

I was happy to see this revised version of the previous manuscript. I appreciate that the authors addressed the major criticisms from the previous review, both mine and those of the other reviewers. The clarifications and added experiments significantly improve and strengthen this study, and I have no additional major comments. This is really a fascinating system and I look forward to seeing continue research along these lines in this and other model systems.

Thank you very much.

Minor comments:

Line 50 – “aggressed” is a bit awkward. Better to say “attacked” or “subjected to aggression by”.

Changed to the latter.

Line 122 – Better to say “before the experiment” rather than “ahead of the experiment”.

Changed (now line 123).

Line 174 – Change “where, as previously shown” to “when, as previously shown”. (and remove comma before “where”).

Changed (now line 176).

Use of commas is inconsistent throughout and would benefit from input from style editor.

We have used the commas the best we could and would be very happy to follow the changes suggested by the style editor.